# An immune checkpoint score system for prognostic evaluation and adjuvant chemotherapy selection in gastric cancer

Jia-Bin Wang[1,2,3,8], Ping Li[1,2,3,8], Xiao-Long Liu [4,8], Qiao-Ling Zheng[5,8], Yu-Bin Ma[6,8], Ya-Jun Zhao[7,8], Jian-Wei Xie[1,2,3], Jian-Xian Lin[1,2,3], Jun Lu[1,2,3], Qi-Yue Chen[1,2,3], Long-Long Cao[1,2,3], Mi Lin[1,2,3], Li-Chao Liu[1,2,3], Ning-Zi Lian[1,2,3], Ying-Hong Yang [5✉], Chang-Ming Huang [1,2,3✉] & Chao-Hui Zheng [1,2,3✉]

Immunosuppressive molecules are extremely valuable prognostic biomarkers across different cancer types. However, the diversity of different immunosuppressive molecules makes it very difficult to accurately predict clinical outcomes based only on a single immunosuppressive molecule. Here, we establish a comprehensive immune scoring system ($ISS_{GC}$) based on 6 immunosuppressive ligands (NECTIN2, CEACAM1, HMGB1, SIGLEC6, CD44, and CD155) using the LASSO method to improve prognostic accuracy and provide an additional selection strategy for adjuvant chemotherapy of gastric cancer (GC). The results show that $ISS_{GC}$ is an independent prognostic factor and a supplement of TNM stage for GC patients, and it can improve their prognosis prediction accuracy; in addition, it can distinguish GC patients with better prognosis from those with high prognostic nutritional index score; furthermore, $ISS_{GC}$ can also be used as a tool to select GC patients who would benefit from adjuvant chemotherapy independent of their TNM stages, MSI status and EBV status.

[1] Department of Gastric Surgery, Fujian Medical University Union Hospital, Fuzhou, China. [2] Key Laboratory of Ministry of Education of Gastrointestinal Cancer, Fujian Medical University, Fuzhou, China. [3] Fujian Key Laboratory of Tumor Microbiology, Fujian Medical University, Fuzhou, China. [4] The United Innovation of Mengchao Hepatobiliary Technology Key Laboratory of Fujian Province, Fuzhou, China. [5] Department of Pathology, Fujian Medical University Union Hospital, Fuzhou, China. [6] Department of Gastrointestinal Surgery, The Affiliated Hospital of Qinghai University, Xining, China. [7] Division of Life Sciences and Medicine, Department of Gastrointestinal Surgery, The First Affiliated Hospital of USTC, University of Science and Technology of China, Hefei, Anhui 230031, China. [8] These authors contributed equally: Jia-Bin Wang, Ping Li, Xiao-Long Liu, Qiao-Ling Zheng, Yu-Bin Ma, Ya-Jun Zhao. ✉email: yyh1555@163.com; hcmlr2002@163.com; wwkzch@163.com

Recently, an increasing number of oncologists have begun to focus their studies on anti-tumour immune responses, which might become fundamental markers in cancer immunotherapy. In recent years, despite remarkable progress in immunotherapy, such as PD1-targeted therapy, there are still a considerable number of patients who cannot benefit from immunotherapy, which may be related to the immunosuppressive environment of tumours. Tumour immunosuppression describes the suppressed host immune responses to tumour antigens, resulting in the reduction or loss of antigens on tumour cells, inhibiting the activation of immune effector cells and the decreased cell viability of cytotoxic T lymphocytes (CTLs) or natural killer cells. Sometimes, tumours develop various tactics to suppress antitumour immunity, leading to the failure of immune regulation of tumour growth[1]. These tactics include expressing a series of receptors on the tumour cell surface, called immune checkpoints or immunosuppressive ligands; e.g., PD-L1, a transmembrane surface antigen with an immunoglobulin-like structure, is distributed in many tissues and interactions with PD-L1 lead to inhibition of T-cell receptor-mediated T-cell activation[2]. Other immune checkpoints (such as FAS-L and IDO) have also been reported to inhibit T-cell responses by depleting tryptophan and producing kynurenine (toxic to lymphocytes) or mediating activation-induced T-cell death[3]. Nevertheless, different types of cancers express diverse immune checkpoints and, even in the same type of tumour, the expression of immune checkpoints is different across patients. For gastric cancer (GC), it is still unknown how many immune checkpoints are expressed and whether they are valuable for predicting the prognosis of GC patients. Thus, testing the expression levels of immune checkpoints in GC patients and using valuable immune checkpoints to form a scoring system will significantly help surgeons accurately perform prognostic assessments.

The immune checkpoints expressed on tumour cells play a key role in protecting tumour cells from attack by host immune responses, especially local immunity[4]. This immunosuppressive tumour microenvironment can assist tumours in escaping immune recognition and thus promote tumour proliferation, local progression and systemic dissemination[5]. Various studies have illustrated that lymphocytes infiltrate into the tumour in many malignant tumours, which represents the local immune response[6–8], and tumour-infiltrating lymphocytes (TILs) are correlated with the prognosis of several types of cancer[9–11]. However, the effect of local immunity on the prognosis of patients can be affected by nutritional status and systemic immune competence. A study indicated that the TIL status was significantly related to the prognostic nutritional index (PNI) score, an indicator of nutritional status and systemic immune competence. Patients with high PNI scores were more likely to have strong lymphocyte infiltration in tumours than those with low PNI scores[12]. Another study has shown that the PNI is associated with the density of $CD4^+$ immune cells, leading to prognostic value for systemic inflammation in GC[13]. Although a high PNI score accompanying strong lymphocyte infiltration in tumours could enhance antitumour immunity, it remains unknown whether a high PNI will benefit patients with an immunosuppressive tumour microenvironment and further validation is still needed. Therefore, a model is needed to evaluate the local immunosuppressive tumour microenvironment and to compare its prognostic value with that of PNI.

Chemotherapy remains an indispensable strategy in the treatment of GC patients, particularly patients with locally advanced, resectable GC[14,15]; however, chemotherapy has two major effects on antitumour immunity. On the one hand, it can kill cancer cells by causing them to elicit an immune response or increasing their susceptibility to immune attack[16]. On the other hand,

chemotherapy can induce immunosuppression by myelosuppression, depleting T lymphocytes and having other immunosuppressive effects[17]. Therefore, determining which patients will benefit from chemotherapy remains a critical problem. Currently, tumor-node-metastasis (TNM) stage is widely used to determine which patients should receive chemotherapy. Obviously, not all patients benefit from chemotherapy when the decision is based on TNM stage. In this situation, other tools are urgently needed to assist TNM staging to determine whether patients are suitable for chemotherapy.

Recently, the molecular The Cancer Genome Atlas (TCGA) classification of GC has been gradually used to predict clinical prognosis beyond TNM staging[18]. Among them, Microsatellite Instability-High (MSI-H)- and Epstein-Barr virus (EBV)-positive GCs are described as unique subgroups of GCs in TCGA with several unique clinicopathologic characteristics, such as abundant TILs, earlier stage and favourable prognosis[19]. However, for Microsatellite Instability-Low (MSI-L)/Microsatellitestability (MSS) or EBV-negative GC patients, there are no practical methods or molecular indicators to evaluate their immune microenvironment status and select corresponding adjuvant therapies.

Immune checkpoints can be alternative choices for predicting prognosis and selecting adjuvant therapies for GC patients, but the accuracy is not satisfactory based on a single molecule. Therefore, a single model is needed to integrate multiple immune checkpoints. To address this requirement, the least absolute shrinkage and selection operator method (LASSO) can be a powerful tool. This is a penalized regression approach that estimates the regression coefficients by maximizing the log-likelihood function (or sum of squared residuals). In addition, it automatically deletes unnecessary covariates, which can provide more features to a regression model with large numbers of covariates[20]. In recent years, the application of the LASSO model has increased, especially in the establishment of models composed of multiple immune checkpoints, and this model can improve the accuracy of predicting prognosis in cancer patients[21].

Therefore, we employ a LASSO Cox regression model to create an immunosuppression scoring system for GC (ISS$_{GC}$), which can serve as a comprehensive score model of multiple immune checkpoints and the systemic immune status (as assessed by the PNI) to improve the prognostic accuracy in GC patients and identify potential beneficiaries of adjuvant chemotherapy. To a certain extent, ISS$_{GC}$ can guide surgeons to develop personalized chemotherapy regimens for GC patients or to choose other treatment regimens, such as immunotherapy. Similarly, it can make the follow-up of GC patients more precise and help to set personalized follow-up schedules.

## Results

**Patient characteristics and ISS$_{GC}$ establishment**. To evaluate the essential immune checkpoints involved in GC, a total of 20 molecules that had been confirmed to perform significant roles in immunosuppression were collected from various manuscripts. The results from analysing the prognostic value of these indicators using immunohistochemistry (IHC) in tissue microarrays (TMAs) showed that only 7 out of the 20 indicators (NECTIN2, CEACAM1, HMGB1, SIGLEC6, ADENOSINE, CD44 and CD155) (Fig. 1 and Supplementary Figs. 4 and 5) showed survival significance ($n = 124$ patients). To validate whether the seven indicators identified from the TMAs were valuable, an expanded immunohistochemical analysis was conducted on gastric tumour samples from 444 patients in our centre, as well as 226 patients from 2 external centres. First, the overall survival rates associated with these seven indicators were evaluated in the patients from our centre. The clinicopathological data are presented in Table 1. The results indicated that the six

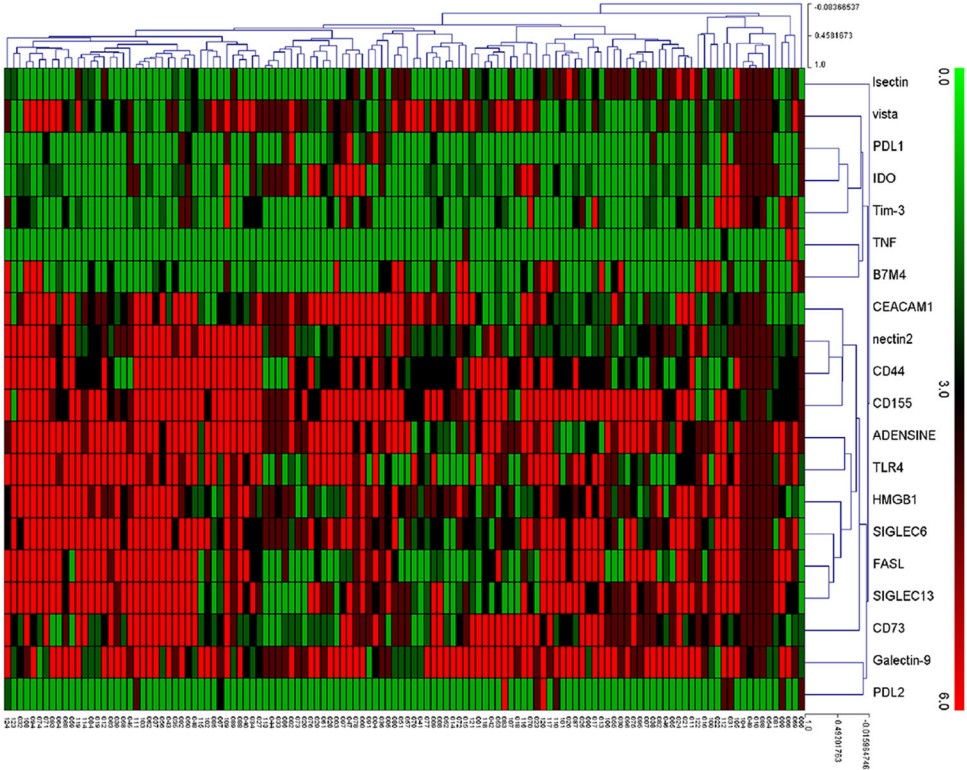

**Fig. 1 A heat map presenting data on 20 immune checkpoints from a TMA containing 124 GC patient samples.** The abscissa represents the patient number of 124 patients in the TMA, and the ordinate represents the expression of LSECtin, VISTA, PD-L1, IDO, TIM-3, TNFRSF14, B7M4, CEACAM1, NECTIN2, CD44, CD155, ADENOSINE, TLR4, HMGB1, SIGLEC6, FAS-L, SIGLEC15, CD73, Galectin-9 and PD-L2.

indicators (NECTIN2, CEACAM1, SIGLEC6, ADENOSINE, CD44 and CD155) were significantly related to the overall survival rate of the patients ($P < 0.05$). Although HMGB1 did not reach statistical significance, it still showed a clear trend that higher expression indicated a worse prognosis (Supplementary Figs. 9 and 10). Considering that immunosuppression may not be caused by a single indicator, the co-function of these indicators should be given more attention. Spearman's correlation analysis showed that there were correlations between the seven indicators (Supplementary Table 3). When the variables were correlated, the traditional Cox model was not applicable, so the LASSO Cox regression model was employed to analyse the seven selected indicators (Supplementary Figs. 11 and 12) and the obtained model formula was as follows: $ISS_{GC} = (0.1095 \times CEACAM1) - (0.0189 \times NECTIN2) - (0.0053 \times HMGB1) + (0.0310 \times CD155) + (0.1237 \times CD44) + (0.0301 \times SIGLEC6) + (0 \times ADENOSINE)$. According to this formula, we eliminated the ADENOSINE index with a coefficient of 0 (less important in this formula). We divided patients into a high-$ISS_{GC}$ group and a low-$ISS_{GC}$ group, and used X-tile to determine the $ISS_{GC}$ cut-off point (1.83) and the PNI cut-off point (46.6) (Supplementary Figs. 1 and 2). To distinguish whether the selected markers were expressed mainly in cancer or immune cells of tissue sections, we performed multi-colour immunofluorescence staining analysis on the six immune indicators with CKpan and CD45 to determine the origin cells of these six indicators in 135 GC patients. The results showed that all six indicators were mostly colocalized with CKpan; therefore, they should mostly originate from GC cells (Fig. 2). Furthermore, Spearman's correlation analysis showed that there was no correlation between CD45 or TIL and six indicators (Supplementary Table 4), which indicated that the six indicators may not relate to the overall immune infiltration or TILs.

As such, the $ISS_{GC}$ was built based on a combination of six immune checkpoints representing the local immunosuppressive tumour microenvironment, which was a vital step to perform subsequent analyses.

**Prognostic value of $ISS_{GC}$: clinicopathological factors and positive marker expression.** To determine whether $ISS_{GC}$ was an independent prognostic factor for the internal cohort, multivariate analysis with a Cox regression model was performed. As summarized in Supplementary Table 5, $ISS_{GC}$ remained a powerful and independent factor after adjustment for clinicopathological prognostic factors among 444 patients (hazard ratio: 2.66, 95% confidence interval 1.97–3.58, $P < 0.001$). The survival analysis showed that the 5-year survival rate of patients with GC in the high-$ISS_{GC}$ group was significantly lower than that in the low-$ISS_{GC}$ group (36.4% vs. 64.3%, $P < 0.001$) (Fig. 3A, B), which meant that the higher the $ISS_{GC}$, the lower the survival rate for patients with GC was. For the six immune checkpoints, patients with higher positive expression of the indicators had poorer prognosis than patients with lower expression of the indicators (P0 vs. P1-2 = 0.239; P0 vs. P3-4 = 0.033; P0 vs. P5-6 = 0.008; P1-2 vs. P3-4 < 0.001; P1-2 vs. P5-6 < 0.001; P3-4 vs. P5-6 = 0.001) (Figs. 3C, D).

Collectively, our research revealed that $ISS_{GC}$ was a strong independent factor. In addition, compared with a single immune checkpoint, a model composed of multiple immune checkpoints could significantly improve the accuracy of prognosis in patients with GC.

**Prognostic value of the $ISS_{GC}$: TNM staging and chemotherapy.** To evaluate the prognostic value of $ISS_{GC}$, Kaplan–Meier analysis and stratification analysis were performed by TNM stage and receipt of chemotherapy. As shown in Fig. 4A–C, after stratification by TNM stage, $ISS_{GC}$ was significantly correlated

**Table 1 Clinical data of internal modelling set.**

| Variable | Training set | |
|---|---|---|
| | **n = 444** | **%** |
| Age(years) | | |
| ≤65 | 249 | 56.1 |
| >65 | 195 | 43.9 |
| Sex | | |
| Female | 109 | 24.5 |
| Male | 335 | 75.5 |
| BMI | | |
| ≤ 25 | 379 | 85.4 |
| > 25 | 65 | 14.6 |
| Surgery type | | |
| Open surgery | 11 | 2.5 |
| Laparoscopic surgery | 433 | 97.5 |
| Resection type | | |
| Part gastrectomy | 197 | 44.4 |
| Total gastrectomy | 247 | 55.6 |
| Tumour size | | |
| ≤ 50 mm | 219 | 49.3 |
| > 50 mm | 225 | 50.7 |
| Pathological type | | |
| Adenocarcinoma | 368 | 82.9 |
| Non-adenocarcinoma | 21 | 4.7 |
| Mix | 55 | 12.4 |
| Grade | | |
| Unknown | 58 | 13.1 |
| High | 6 | 1.4 |
| Middle | 155 | 34.9 |
| Low | 161 | 36.3 |
| Mix | 64 | 14.4 |
| Number of lymph node examined | | |
| ≤ 15 | 19 | 4.3 |
| > 15 | 425 | 95.7 |
| pT | | |
| T1 | 47 | 10.6 |
| T2 | 53 | 11.9 |
| T3 | 168 | 37.8 |
| T4 | 176 | 39.6 |
| pN | | |
| N0 | 112 | 25.2 |
| N1 | 90 | 20.3 |
| N2 | 85 | 19.1 |
| N3 | 157 | 35.4 |
| AJCC7th | | |
| I | 63 | 14.2 |
| II | 138 | 31.1 |
| III | 243 | 54.7 |
| Adjuvant chemotherapy | | |
| No | 237 | 53.4 |
| Yes | 207 | 46.6 |
| PNI | | |
| ≤ 46.6 | 191 | 43 |
| > 46.6 | 253 | 57 |
| MSI status | | |
| MSI-H | 111 | 25.0 |
| MSI-L/MSS | 333 | 75.0 |
| EBV status | | |
| Negative | 423 | 95.3 |
| Positive | 21 | 4.7 |

with prognosis: a higher $ISS_{GC}$ indicated a worse prognosis in any stage, which meant that $ISS_{GC}$ could distinguish patients with poor prognosis from those with disease even in an early TNM stage. Compared with patients in the high-$ISS_{GC}$ group, patients in the low-$ISS_{GC}$ group who received chemotherapy had a better prognosis ($P = 0.003$). Furthermore, the stratification analysis

showed that patients with stage II or III disease in the low-$ISS_{GC}$ group had a better prognosis when they received chemotherapy than when they did not, whereas patients in the high-$ISS_{GC}$ group had no significant difference in prognosis with or without chemotherapy (Fig. 4D–I). Therefore, our research revealed that the benefits of chemotherapy were limited to stage II or III patients from the low-$ISS_{GC}$ group.

In summary, $ISS_{GC}$ was able to re-stratify the risk of patients with different TNM stages. This demonstrates that combining the TNM stage with $ISS_{GC}$ can better assist surgeons in identifying and classifying patients with potential risk factors, which increases the prognostic value of TNM staging. In addition, according to our research, chemotherapy should be recommended for patients in the low-$ISS_{GC}$ group with TNM stage II or III disease. However, patients in the high-$ISS_{GC}$ group with TNM stage II or III disease derive poorer effects from chemotherapy than patients in the low-$ISS_{GC}$ group and other treatment options should be further considered.

**Prognostic value of the $ISS_{GC}$: MSI- and EBV-associated subtypes.** To clarify whether the MSI- or EBV-associated subtypes would affect the prognostic value of our $ISS_{GC}$, we determined the MSI status and EBV status of each case. Kaplan–Meier analysis and stratification analysis were performed to evaluate the prognostic value of $ISS_{GC}$ after stratification by MSI- or EBV-associated subtypes. As shown in Fig. 5A–D, after stratification by MSI subtypes or EBV-associated subtypes, the $ISS_{GC}$ was still significantly correlated with the prognosis of all subtypes: a higher $ISS_{GC}$ indicated a worse prognosis independent of MSI and EBV status, indicating that the prognostic value of the $ISS_{GC}$ was not needed to consider the MSI and EBV status ($P < 0.001$). After stratification by MSI subtypes and receipt of chemotherapy, Kaplan–Meier and stratification analysis showed that patients in the low-$ISS_{GC}$ group either with MSI-H or MSI-L/MSS all had a better prognosis when they received chemotherapy, whereas it seems that the high-$ISS_{GC}$ group had no additional significant benefit regarding overall survival with chemotherapy in both the MSI-H and MSI-L groups (Fig. 5E–H). Similarly, after stratification by EBV-associated subtypes and receipt of chemotherapy, we also found that the low-$ISS_{GC}$ group of EBV-negative patients could benefit from adjuvant chemotherapy and had a better prognosis than those without chemotherapy, whereas the high-$ISS_{GC}$ group of EBV-negative patients could not benefit from adjuvant chemotherapy and had no significant difference in prognosis regardless of whether they received chemotherapy (Fig. 5I, J). However, we could not analyse the effects of chemotherapy on the prognosis of EBV-positive GC patients due to the small number of patients.

In summary, the $ISS_{GC}$ scoring system was able to distinguish patients with poor prognosis and screen patients who might benefit from adjuvant chemotherapy independent of their MSI or EBV status and could further provide additional immune microenvironment characterization among MSI-L/MSS and EBV-negative GC patients.

In addition, we found that the $ISS_{GC}$ + TNM staging + MSI status in the internal cohort was better than TNM staging + MSI status by the C-index ($ISS_{GC}$ + TNM + MSI = 0.705 (0.670–0.739) vs. TNM + MSI = 0.640 (0.603–0.676), $P < 0.001$) and AIC (ISS + TNM + MSI) = 2254.93 < AIC (TNM + MSI) = 2300.13). It showed that the TNM stage + MSI status combined with our $ISS_{GC}$ would be more accurate when assessing the prognostic value of GC patients.

**Prognostic value of the $ISS_{GC}$: combination with the PNI.** To quantify the nutritional and immune status of patients, the PNI

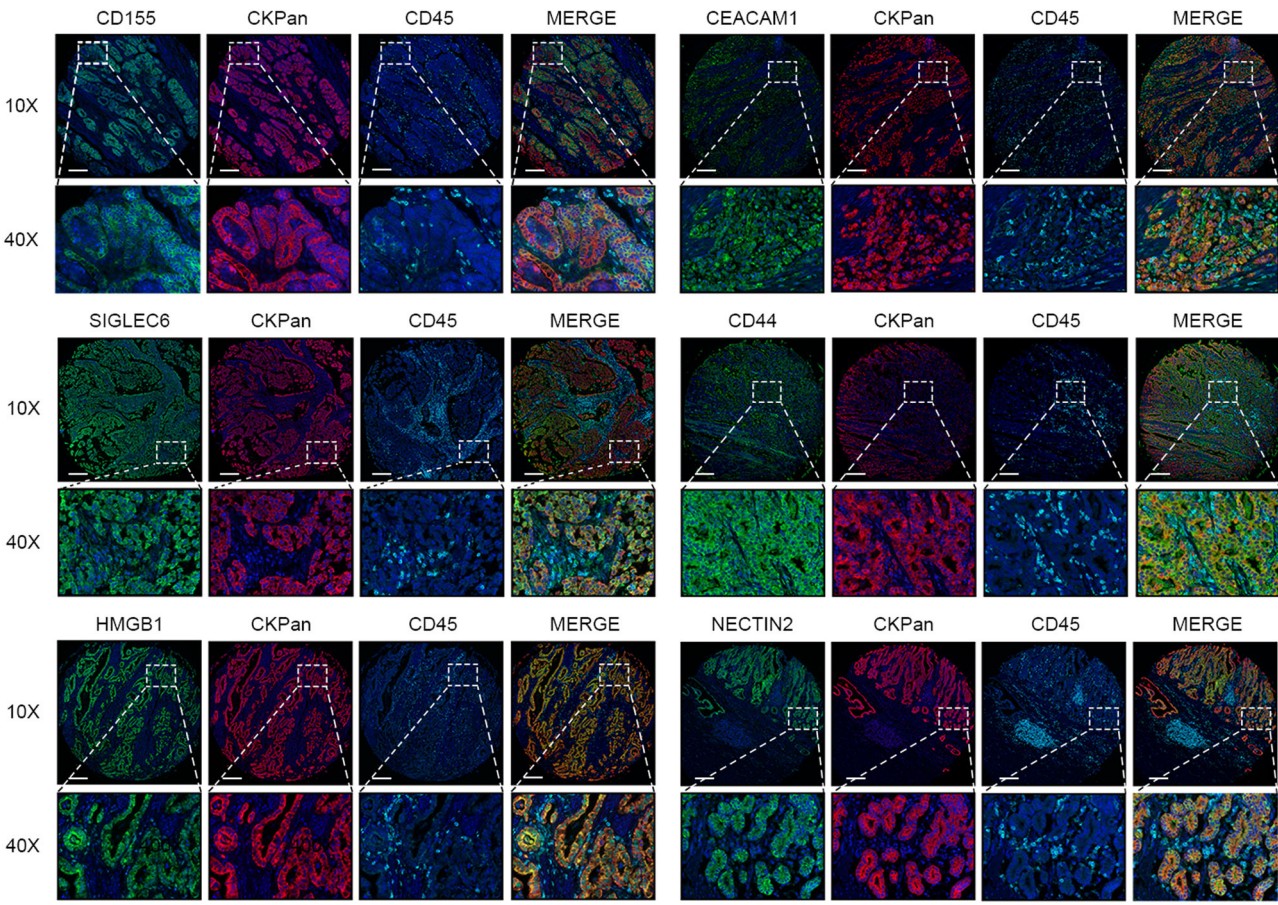

**Fig. 2 The co-localization of 6 indicators (CD155, NECTIN2, CEACAM1, HMGB1, SIGLEC6, CD44), CD45 and gastric cancer CKpan.** Multi-colour immunofluorescence staining of anti-CKpan and CD45 antibodies with CD155, NECTIN2, CEACAM1, HMGB1, SIGLEC6 or CD44 antibodies. Scale bar = 200 μm.

was employed, which can be easily and conventionally measured. When assessing the relationship between $ISS_{GC}$ and the PNI, a negative correlation was found in the internal cohort (Supplementary Table 6 and Fig. 6A), which indicated that patients with a more immunosuppressive tumour microenvironment possibly had a lower systemic immunity status. In addition, our research showed that not all patients with a high PNI value (PNI > 46.6) could benefit and only when patients had both a high PNI (PNI > 46.6) and a low $ISS_{GC}$ ($ISS_{GC} \leq 1.83$) did they have a better prognosis (Fig. 6B). This phenomenon may be because the overall prognosis of patients with high $ISS_{GC}$ values ($ISS_{GC} > 1.83$) was worse than that of patients with low $ISS_{GC}$ values ($ISS_{GC} \leq 1.83$).

Above all, our results showed that the PNI and $ISS_{GC}$ had a significant effect on survival and stratification based on the PNI and $ISS_{GC}$ can better distinguish patients with poor prognosis, whereas this ability was limited in analysis that only used the PNI.

**External validation**. To confirm whether $ISS_{GC}$ had the same excellent prognostic value in different populations, we further applied it to an external validation cohort and found similar results (Supplementary Table 1 and Supplementary Fig. 14). The survival analysis showed that the 5-year survival rate of GC patients in the high-$ISS_{GC}$ group was significantly lower than that in the low-$ISS_{GC}$ group ($P < 0.001$) (Fig. 7A). In terms of prognostication (C-index), the combination of $ISS_{GC}$ + TNM staging + MSI status in the external validation cohort was also better than TNM staging + MSI status ($ISS_{GC}$ + TNM + MSI = 0.728 (0.677–0.778) vs. TNM + MSI = 0.705 (0.655–0.755), $P < 0.001$) (AIC (ISS + TNM + MSI) = 855.23 < AIC (TNM + MSI) = 861.09). After stratification by TNM stage, MSI

status, or EBV status, $ISS_{GC}$ was significantly correlated with prognosis: the higher the $ISS_{GC}$, the worse the prognoses of patients (Figs. 7B–D and 8A–D). Further stratification analysis showed that it induced better outcomes in patients with chemotherapy than without chemotherapy in the low-$ISS_{GC}$ group, whereas there was no significant difference in survival between the groups with chemotherapy and without chemotherapy in the high-$ISS_{GC}$ group of GC patients (Figs. 7E–J and 8E–J). A negative correlation between the $ISS_{GC}$ and PNI still existed in the external cohort, and when the $ISS_{GC}$ was high, even a high PNI was unable to predict a good prognosis for GC patients (Supplementary Fig. 15).

Overall, the scoring system introduced here works well in predicting the prognosis of GC patients, as assessed in two distributed regions of China, which means that it may perform well throughout China. It may guide surgeons in administering personalized chemotherapy or other treatments in GC patients.

**Discussion**

Despite the use of multimodal treatments, including neoadjuvant chemotherapy, surgery, postoperative radiotherapy and chemotherapy, the overall prognosis of GC patients is still unsatisfactory[22]. Therefore, exploring the best individual treatment for GC patients has become a hot topic. In recent years, more attention has been paid to the role of immune inhibitory molecules expressed by tumour cells, such as PD-L1, in mediating tumour progression[23–26]. Overall, the positive rate of PD-L1 expression in GC was only 14.3–29.6%[27–29] and the positive rate of PD-L1 was <10% in our centre (data not shown). Furthermore, not all patients with positive expression of PD-L1 can benefit

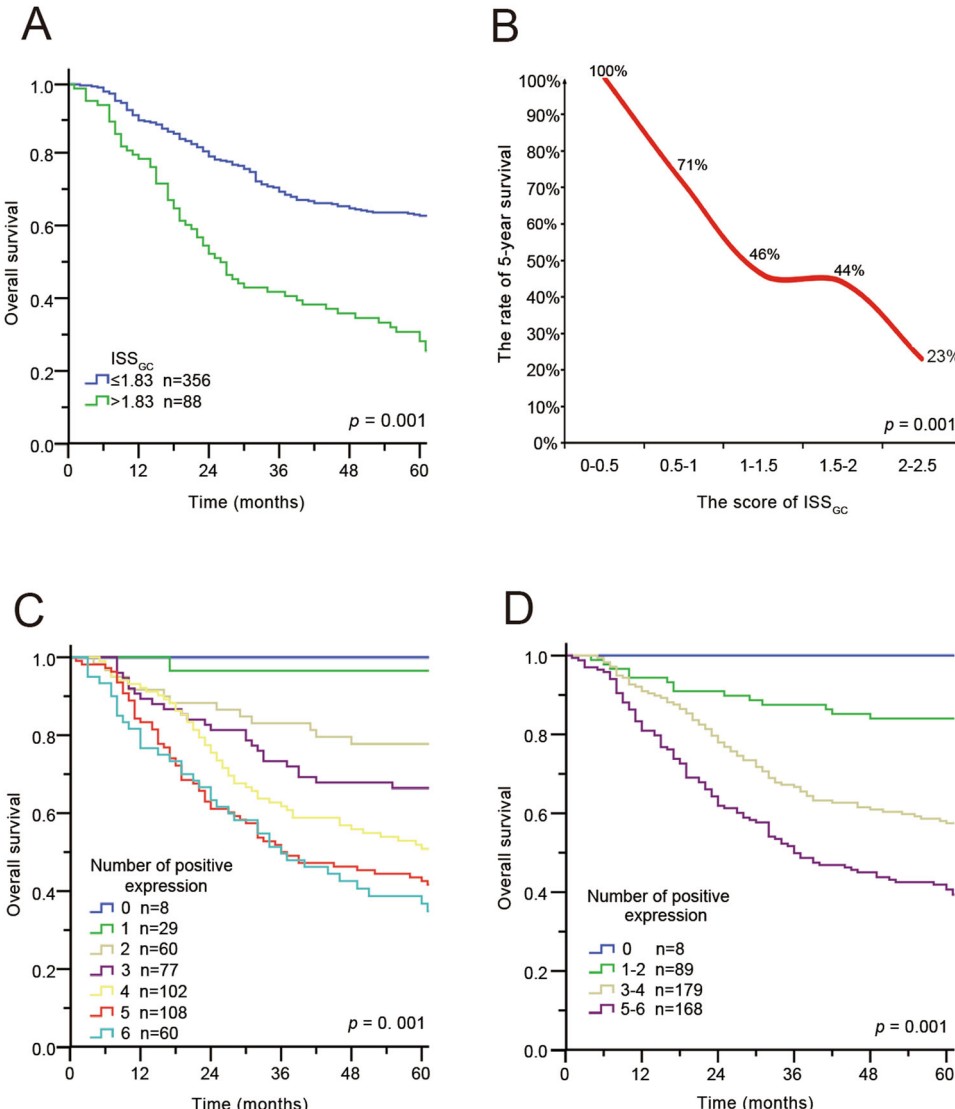

**Fig. 3 Prognostic value of ISS_GC. A** Kaplan–Meier curves for overall survival in GC patients according to the ISS_GC. *p*-Values have been calculated using the log-rank test. **B** Negative correlation between survival rate and ISS_GC. Statistical significance was determined by a two-tailed paired Student's *t*-test. **C**, **D** Kaplan–Meier curves for overall survival in GC patients according to the number of positive ligands. *p*-Values for all survival analyses have been calculated using the log-rank test.

from PD-L1 inhibitors, including patients with GC[23,30–32]. Thus, many scholars hypothesized that in addition to PD-L1, there may be other co-expressed immune checkpoints on tumour cells that may play a role alone or in combination with PD-L1 to down-regulate antitumour immunity. Therefore, 20 molecules that may mediate tumour immunosuppression were collected from the literature among different tumours in the current study. Most of these molecules play important roles in inhibiting T-cell responses, producing toxins to lymphocytes or mediating activation-induced T-cell death. It has also been reported that they mediate immunosuppression by regulating the activity of CTLs. Mechanistically, CD44 reduced the sensitivity of tumour cells to CTLs by downregulating the Fas-FasL pathway, causing tumours to escape CTL killing; CEACAM1 affected the immune tolerance of T cells through the TIM-3 signalling pathway and suppressed the immune response of T cells to cancer cells; both SIGLEC6 and HMGB1 played an immunosuppressive function on CTLs by regulating the activity of mast cells; and CD155 and NECTIN2 are both members of the Nectin-like molecule family, which affect the activity of CTLs, thereby inhibiting antitumour

immunity (Supplementary Table 2). As is shown in the multi-colour immunofluorescence staining analysis of 135 GC tissues, ~90% of tissues show that NECTIN2, CEACAM1, HMGB1, SIGLEC6 and CD15 were expressed in tumour cells, and CD44 was expressed in tumour cells in about 70% of tissues. However, whether these molecules play the same roles in GC as they do in other tumours remain unclear. It is also unknown whether combined targeting of these molecules can lead to a better prognosis for GC patients. Thus, ISS_GC, a simple and practical way to assess these molecules, was established by LASSO Cox regression to serve as an indicator of the tumour immunosuppression level.

Accurate prognostic assessment is an important prerequisite for selecting appropriate treatment options. TNM stage is essential to assess prognosis and to develop a treatment strategy. However, these decisions are mostly based on tumour char-acteristics. ISS_GC plays a bioimmunological role and is able to provide further information beyond the TNM stage. It also pro-vides the possibility for patients with the same TNM stage to receive different treatments to improve their long-term prognosis.

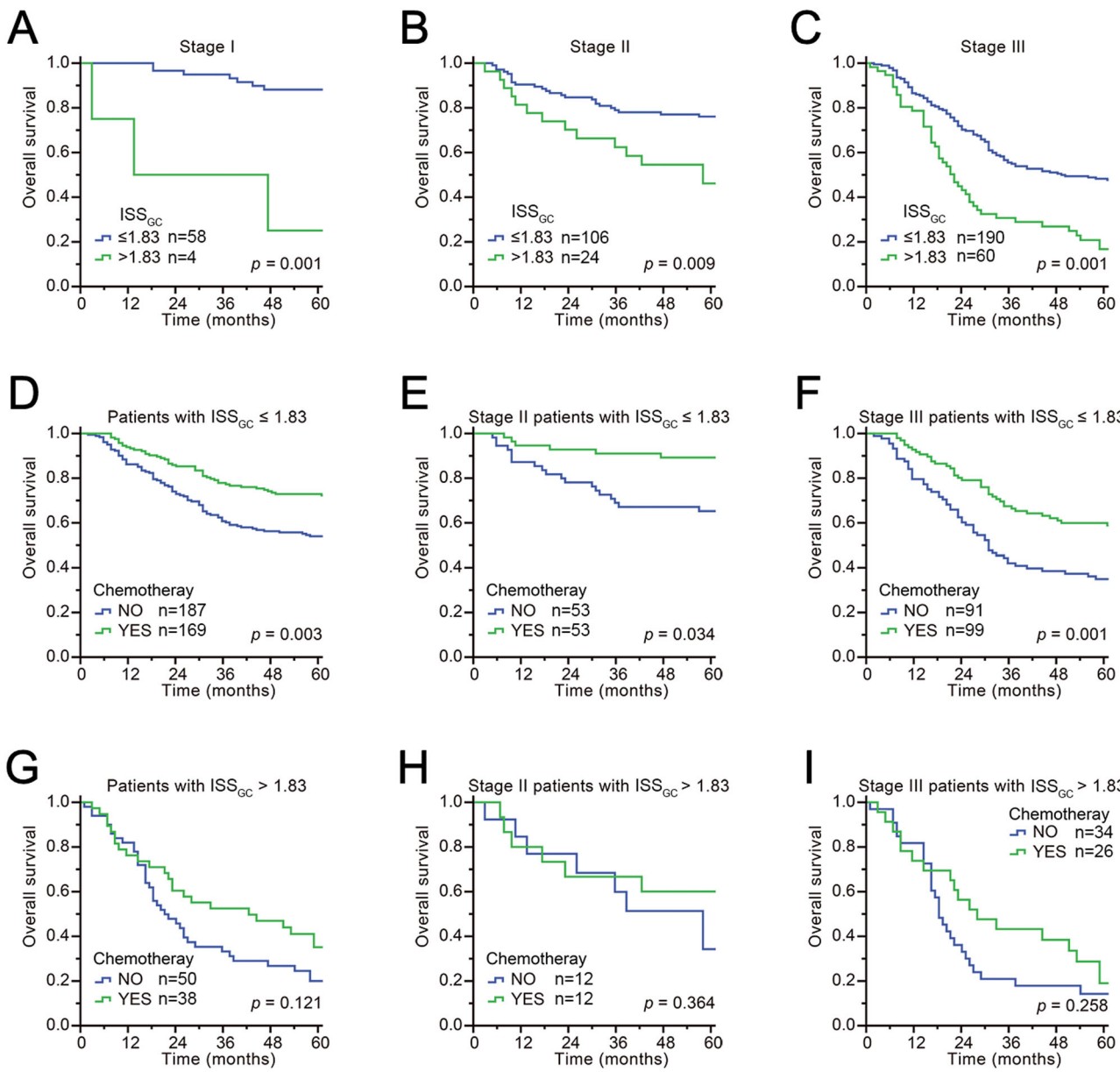

**Fig. 4 Prognostic value of TNM stage and receipt of chemotherapy according to ISS$_{GC}$ stratification. A–C** Overall survival based on TNM stage in GC patients after ISS$_{GC}$ stratification is presented using Kaplan–Meier curves. **D–I** Overall survival analysis of chemotherapy and TNM stage combined in GC patients after ISS$_{GC}$ stratification is presented using Kaplan–Meier curves. *p*-Values for all survival analyses have been calculated using the log-rank test.

Adjuvant chemotherapy is a standard component of therapies for patients with resectable stage II or III GC and improves their outcomes[14,33,34]. The ISS$_{GC}$ showed an ability to predict sensitivity to chemotherapy. The mechanism(s) of immune responses to GC adjuvant chemotherapy have not been thoroughly elucidated[19]. However, accumulating evidence indicates that the efficacy of conventional chemotherapy not only involves direct cytostatic/cytotoxic effects but is also influenced by the (re)activation of tumour-targeting immune responses[35,36]. Chemotherapy can increase the immunogenicity of malignant cells or inhibit the immunosuppressive circuitries that are established by developing neoplasms[35]. For residual cancer cells (those that fail to be killed by chemotherapy) or remaining micrometastases in a stage of dormancy, one effective way to kill these tumour cells was based on stimulating anticancer immune responses[37]. Therefore, based on our index and without any other relevant published studies, 5-fluorouracil-based chemotherapy strategies may not be

suitable for patients with stage II and III disease, and the chemotherapy regimen should be changed to achieve better results. Although TNM staging is commonly applied to prognosis prediction and treatment guidance of patients with GC in clinical diagnosis and treatment, the molecular TCGA classification of GC has also gradually been used. For example, the pembrolizumab has been approved by Food and Drug Administration for the treatment of patients with unresectable or metastatic and MSI-H GCs[19]. Furthermore, some studies also showed that EBV-positive GCs were effective for the treatment of Avelumab and other immune checkpoint drugs[38–40]. However, patients with MSI-L/MSS or EBV-negative GCs had no other optimal treatment choices[41]. Considering the differences in the immune microenvironment between MSI-H and MSI-L/MSS tumours or EBV-positive and EBV-negative tumours, we wished to create a model for prognosis prediction and adjuvant chemotherapy selection that was not affected by patients' MSI or EBV status and

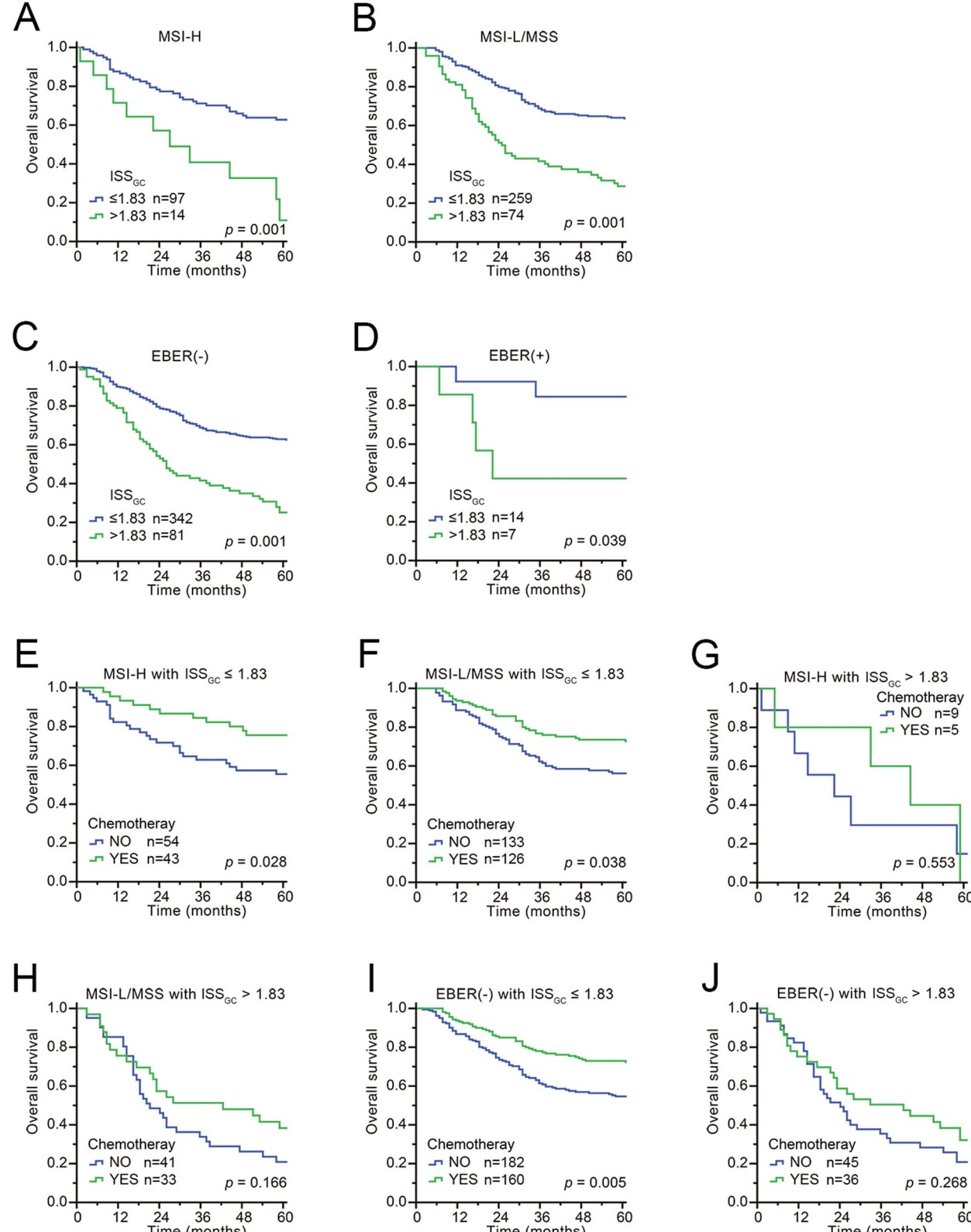

**Fig. 5 Prognostic value of ISS$_{GC}$ score on gastric cancer patients and receipt of adjuvant chemotherapy stratified by their MSI or EBV status. A, B** Overall survival based on MSI status in GC patients after ISS$_{GC}$ stratification is presented using Kaplan–Meier curves. **C, D** Overall survival based on EBV status in GC patients after ISS$_{GC}$ stratification is presented using Kaplan–Meier curves. **E–H** Overall survival analysis of chemotherapy based on MSI status in GC patients after ISS$_{GC}$ stratification is presented using Kaplan–Meier curves. **I–J** Overall survival analysis of chemotherapy in EBV-negative GC patients after ISS$_{GC}$ stratification is presented using Kaplan–Meier curves. p-Values for all survival analyses have been calculated using the log-rank test.

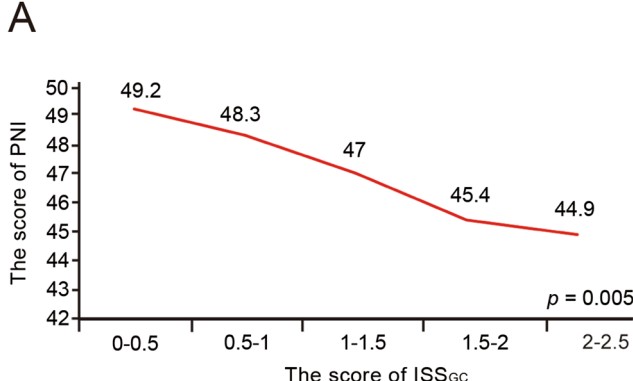

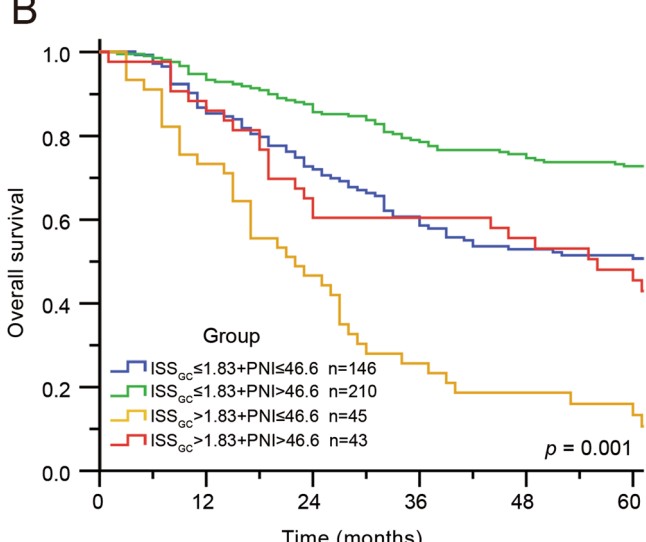

**Fig. 6 The PNI is related to the ISS$_{GC}$. A** Negative correlation between the PNI and the ISS$_{GC}$. Statistical significance was determined by a two-tailed paired Student's $t$-test. **B** Four groups were created according to the cut-off points of the ISS$_{GC}$ and the PNI, and the overall survival in GC patients of these four groups was evaluated by Kaplan–Meier curves. $p$-Values for survival analyses have been calculated using the log-rank test.

provide additional immune microenvironment characterization among MSI-L/MSS and EBV-negative GCs. Therefore, we hope to be able to improve the efficacy of chemotherapy in patients with GC by combining the model we built with individual immunosuppressive agents.

The PNI quantifies the nutritional and immunological statuses, which can be measured easily from clinical tests[12,42]. It was initially designed to evaluate preoperative nutritional conditions and surgical complications in patients with gastrointestinal cancers[43]. Since then, the significance of the PNI as a prognostic predictor has been revealed in various types of human cancers[44–47]. Okadome et al.[12] confirmed the relationship between the PNI and local tumour immunological response, in which a high PNI was the factor promoting good prognosis in patients with oesophageal cancer. That study revealed that the PNI affected the prognosis via local immune responses. This is similar to our results. Therefore, we believe that local immunosuppression influences antitumour functions via the immune cell-mediated systemic immune state[48–50].

Collectively, our study provides a method to investigate the prognostic effects of local immunosuppression by analysing a combination of immune checkpoints. After stratification by ISS$_{GC}$, the immune infiltration score was able to distinguish

patients with poor prognosis of the low-ISS$_{GC}$ group patients (Supplementary Fig. 16). Combined use of the ISS$_{GC}$ with the TNM stage was able to improve the prognostic accuracy to beyond that achieved by TNM staging alone. Moreover, the ISS$_{GC}$ also assisted in assessing the benefits of chemotherapy. Nevertheless, ISS$_{GC}$ is applicable only in patients with GC. When applying this scoring system to other types of malignancies, we believe that the immune checkpoints should be reselected and recombined, mainly due to the diversity of molecules expressed in different cancers. Our study also has some limitations. First, this study is a retrospective study from the Chinese population. Because of the genetic differences between races, which may affect the overall detection rate of MSI-H in GC patients, further validation of whether ISS$_{GC}$ is appropriate for other populations is needed. Second, the immune parameters of this study were obtained from the literature in which they were reported; as such, the latest discovered molecules may not have been included. Third, the number of patients with ISS$_{GC}$ > 1.83 was relatively small, which may affect the accuracy of the results. There were too few patients with EBV-positive GCs to analyse the effects of chemotherapy on their prognosis, which was also one of the deficiencies of this article. Of course, a prospective, multi-centre clinical trial is essential to validate our findings.

## Methods

**Patient and tissue specimens**. This study included a total of 794 GC tissues collected from January 2010 to April 2014 at Fujian Medical University Union Hospital, Affiliated Hospital of Qinghai University, and the First Affiliated Hospital of University of Science and Technology of China. To screen for valuable indicators, TMAs were used, which consisted of 124 patients from Fujian Medical University Union Hospital. Then, 444 patients from Fujian Medical University Union Hospital were included to establish the ISS$_{GC}$; 226 patients from two external centres, Qinghai University Hospital (118 patients) and the First Affiliated Hospital of University of Science and Technology of China (108 patients), were employed for external validation cohorts. Gastric tissue specimens included tumour tissues of the stomach and adjacent non-tumour tissues, which were embedded in paraffin for immunohistochemistry. The inclusion criteria were as follows: (a) histological identification of GC; (b) no other malignant tumours or distant metastases; (c) availability of follow-up data and clinicopathological characteristics; and (d) TNM staging of GC tumours according to the 2010 International Union Against Cancer guidelines. The exclusion criteria were as follows: (1) death within 1 month after the operation and (2) patients who received chemotherapy or radiotherapy before surgery. All participating patients with advanced GC routinely received fluorine-based chemotherapy. The study was approved by the Ethics Committees of Fujian Medical University Union Hospital, the Affiliated Hospital of Qinghai University, and the First Affiliated Hospital of University of Science and Technology of China. This study has been approved by the Ethics Committee of Union Hospital Affiliated to Fujian Medical University (Ethics approval number of scientific research project: 2020KY034). Informed consent was obtained from all participants.

**Tissue microarray**. A total of 124 GC tissue samples were selected from January 2013 to April 2014 (Supplementary data 1). The pathologists marked the paraffin specimens according to the tumour position of the haematoxylin and eosin-stained sections and immunohistochemical slides. The areas with more tumour tissue were selected and labelled. No representative areas of necrotic and haemorrhagic materials were selected to prepare tissue chips for experiments. Paraffin and an equal amount of beeswax were mixed to make two blank wax blocks. A 1 mm diameter puncture hole was created in the blank paraffin to separate 2 holes and 80 punches were made. Each patient specimen had two 1.5 mm tissue cores. Then, the tumour-labelled wax block was sampled with a tissue analyser, the sampled tissue was placed into the corresponding channel of the blank wax block and the determined array position was transferred to the recipient paraffin block. The prepared tissue chip wax block was placed in a 60 °C incubator for 30 min to soften the paraffin, compressed to fully set, allowed to cool at room temperature overnight and placed in the refrigerator as an experimental reserve. The wax block was sliced repeatedly at a thickness of 2 μm. After slicing, the slices were placed in a 37 °C electric heating oven to bake, cooled at room temperature, placed in a refrigerator storage, and removed to be used during experiments.

**Clinicopathological data**. The clinicopathological characteristics of the patients were obtained retrospectively from medical records. These characteristics included age, sex, body mass index, surgery type, resection type, tumour size, pathological type, number of examined lymph nodes, adjuvant chemotherapy and overall

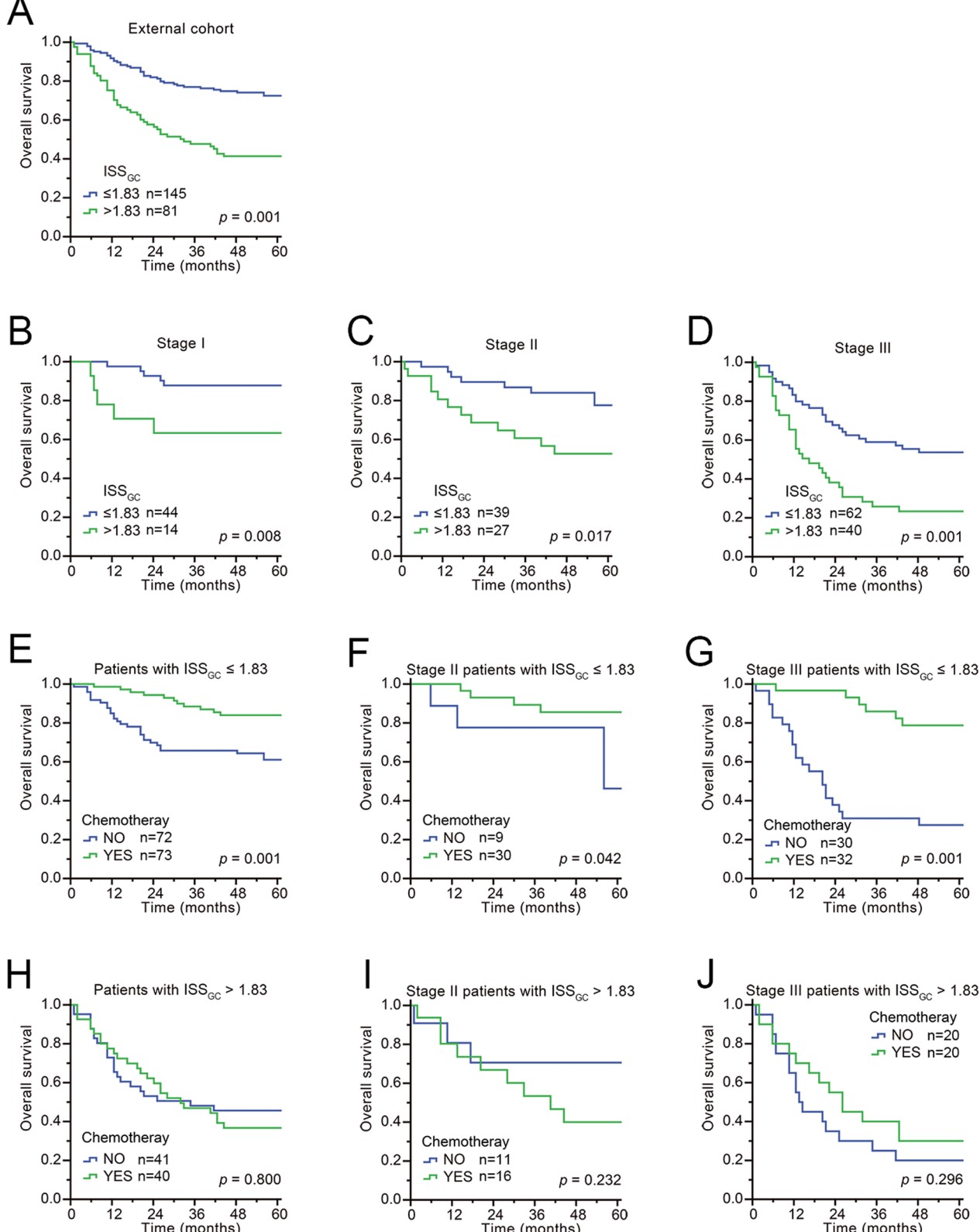

**Fig. 7 External verification. A** The overall survival according to the ISS$_{GC}$ was validated in external cohorts using Kaplan–Meier curves. **B**–**D** The overall survival according to TNM stage after stratification by ISS$_{GC}$ in patients from external centres was similar to that in patients from our centre. **E**–**J** The overall survival based on receipt of chemotherapy and TNM stage after stratification according to the ISS$_{GC}$ in patients from external centres was similar to that in patients from our centre. *p*-Values for survival analyses have been calculated using the log-rank test.

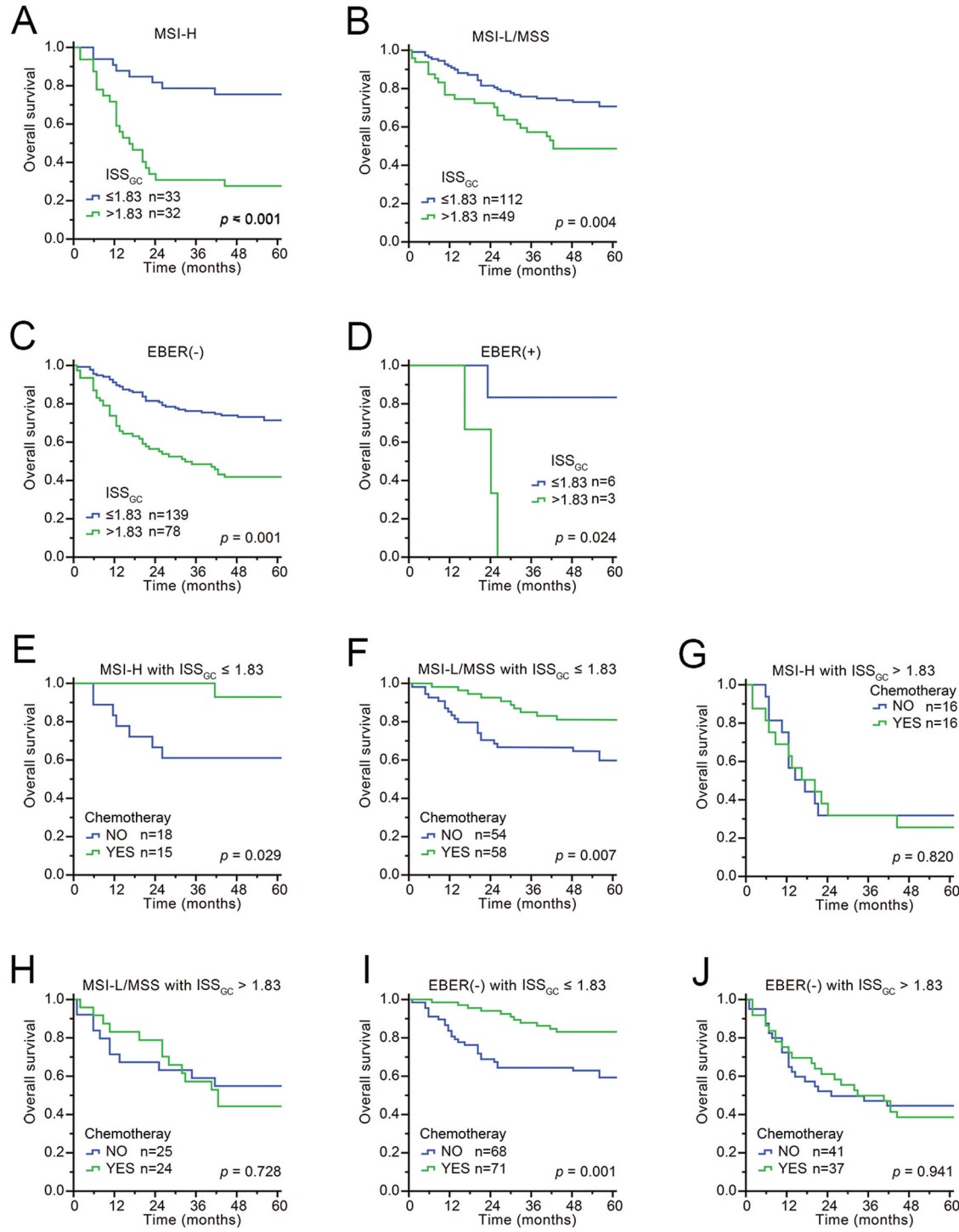

**Fig. 8 External validation.** Prognostic value of ISS$_{GC}$ score on gastric cancer patients and receipt of adjuvant chemotherapy stratified by their MSI or EBV status. **A**, **B** Overall survival based on MSI status in GC patients after ISS$_{GC}$ stratification is presented using Kaplan–Meier curves. **C**, **D** Overall survival based on EBV status in GC patients after ISS$_{GC}$ stratification is presented using Kaplan–Meier curves. **E**–**H** Overall survival analysis of chemotherapy based on MSI status in GC patients after ISS$_{GC}$ stratification is presented using Kaplan–Meier curves. **I**, **J** Overall survival analysis of chemotherapy in EBV-negative GC patients after ISS$_{GC}$ stratification is presented using Kaplan–Meier curves. $p$-Values for survival analyses have been calculated using the log-rank test.

survival (OS). The three centres were individually staffed to collect, organize, enter, and follow up the data in accordance with the provisions of the Statute and performed unified maintenance on the database on a regular basis. The expression and score of each immunosuppression indicator were evaluated by the same two pathologists according to a unified scoring standard to exclude the result bias caused by measurement errors. At the same time, the key clinical indicators of the three centres, such as the patient's tumour pathology type, TNM stage, postoperative

chemotherapy status, and survival status, were all checked by the dedicated staff to minimize deviation. The data of 444 patients in the internal centre of the modelling group were relatively complete, with no missing data (Table 1 and Supplementary Data 2). The adjuvant chemotherapy, survival, TNM stage, age, gender, tumour size and PNI index of 226 patients in the external centre of the validation group were also complete (Supplementary Table 1 and Supplementary Data 3). For patients in the external validation group who lost the continuous variable of body mass index

data, the average value was substituted in this study and it did not affect the validation of the $ISS_{GC}$. The PNI was calculated as $10 \times$ serum albumin (g/dL) + $0.005 \times$ total lymphocyte count (per mm$^3$). We used X-tile software to select the best cut-off value for the PNI. The cut-off for the PNI was 46.6 in the current study (Supplementary Fig. 1).

**Follow-up**. Follow-up was performed every 3 months in the first year and every 6 months after the second year. All surviving patients were followed for more than 5 years. We defined overall survival as the time from surgery to the time of the last follow-up (April 2019), the time of death or the database deadline (time lost to follow-up). The survival time of deceased patients was defined as the time from surgery to the time of death and the survival time of surviving patients was defined as the time from surgery to the time of the last follow-up. Our follow-up method was conducted by clinicians of the three centres in accordance with the unified standards of the Japanese Statute, and the overall loss of follow-up rate was 4.4%.

**Statistical analysis**. All data were processed using SPSS 20.0 (SPSS, Inc., Chicago, IL) and R software (version 4.0.0). For continuous variables, the $t$-test or Mann–Whitney $U$-test was used. We used the $\chi^2$-test or Fisher's exact test to compare categorical variables of clinical characteristics. Correlation analysis was performed by the Spearman test or Kruskal–Wallis test and we used the Kaplan–Meier method to estimate median survival. The association of relevant clinicopathological variables with OS was assessed using the Cox proportional hazard model. Stepwise backward variable removal was applied to a multivariate model to identify the most accurate and minimal set of predictors. Clustering charts were used to describe the level of protein expression in each case. A stepwise regression method was used in the relevant clinical pathology variable analysis. The clinical pathology data were first subjected to single-factor Cox analysis and then the selected meaningful variables were included in the multivariate regression analysis. The inclusion criterion for stepwise regression analysis was Pe = 0.05 and the elimination criterion was Pr = 0.1. X-tile software (version 3.6.1) was used to select the best cut-off value for the $ISS_{GC}$ (the cut-off of $ISS_{GC}$ was 1.83) (Supplementary Fig. 2). After 20 kinds of immune checkpoint indexes were stratified by X-tile, the prognostic efficacy was analysed using the Kaplan–Meier method (two-sided statistical tests). Seven meaningful indexes (HMGB1, NECTIN2, SIGLEC6, CEACAM1, ADENOSINE, CD44 and CD155) were identified, and they were incorporated into the LASSO regression (kernel Cox) model to construct the $ISS_{GC}$ classifier. The LASSO model compresses the coefficients of some meaningless variables to 0. It can eliminate offsets caused by correlations between variables, resulting in a more stable scientific model. The data were analysed using EmpowerR software (version 2.0). The C-index software package and AIC software package of R software (version 4.0.0) were used to evaluate and compare the model's ability to predict survival. The C-index software package provides an inverse of the probability of the censoring weigthed estimate of the concordance probability to adjust for right censoring. The censoring when calculating the C-index in this study was right censoring, including patients who were lost to follow-up (a total of 670 patients were included, 18 were lost to follow-up) and patients who were still alive by the time of the database. We defined the survival time of patients who were lost to follow-up as the time from surgery to the last follow-up time and the survival time of patients who were still alive in the end was defined as the time from surgery to the database deadline. All $P$-values < 0.05 were considered significant differences.

**Immunohistochemistry**. Based on the collection and review of relevant literature (Supplementary Table 2), we selected 20 immune checkpoints for immunohisto-chemical staining analysis: CD73 (ab175396, Abcam, 1 : 200), Galectin-9 (54330 S, Cell Signaling Technology (CST), 1 : 800), HMGB1 (ab18256, Abcam, 1 : 1000), FAS-L (ab186671, Abcam, 1 : 200), SIGLEC6 (ab38581, Abcam, 1 : 200), SIGLEC15 (ab174723, Abcam, 1 : 200), TLR4 (ab13556, Abcam, 1 : 200), ADENOSINE (ab40002, Abcam, 1 : 250), CEACAM1 (44464S, CST, 1 : 400), NECTIN2 (95333S, CST, 1 : 100), CD44 (3570S, CST, 1 : 50), CD155 (81254 S, CST, 1 : 200), VISTA (54979S, CST. 1 : 300), IDO (86630S, CST, 1 : 400), LSECtin (ab181196, Abcam, 1 : 250), PD-L2 (82723S, CST, 1 : 200), TNFRSF14 (ab89479, Abcam, 1 : 100), PD-L1 (13684S, CST, 1 : 200), TIM-3 (45208S, CST, 1 : 400), and B7M4 (ab209242, Abcam, 1 : 1000) (Supplementary Figs. 3–5). The mean percentage and intensity of positive cells in five randomly selected fields were evaluated to represent protein expression levels. The scoring criteria (Supplementary Fig. 6) were as follows: the staining intensity was categorized as 0 (no staining), 1 (weak staining, light yellow), 2 (medium staining, yellow-brown) or 3 (strong staining, brown), and the proportion of positive staining tumour cells was categorized as 0 (≤5% positive cells), 1 (6–25% positive cells), 2 (26–50% positive cells) or 3 (≥51% positive cells). The final expression was calculated by multiplying the staining intensity score by the proportional staining score (total from 0 to 9). Patients with a final score of <4 were classified as the low expression group, and patients with a score ≥ 4 were classified as the high expression group.

We performed CD45 (ab10558, Abcam, 1 : 200), CD3 (ab16669, Abcam, 1 : 150) and CD8 (ab4055, Abcam, 1 : 200) immunohistochemical staining on the tumour tissue of 444 patients to evaluate the overall immune infiltration and TIL infiltration. The expression of CD45 reflects overall immune infiltration. We evaluated CD45 cell infiltration and counted positively stained cells in each tumour region under ×400 magnification (the mean percentage of positive cells in five fields was analysed) and the scoring standard was 0 points for <5% CD45-positive cells, 1 point for 5–25% CD45-positive cells, 2 points for 26–50% CD45-positive cells and 3 points for >50% CD45-positive cells, as shown in Supplementary Fig. 7A. To date, there have been no reports on how to evaluate TIL infiltration in GC by IHC, so we adopted and modified the GALON scoring, which used CD3 and CD8 as markers for reflecting the condition of TIL[51]. The tumour area was divided into the centre of the tumour (CT) and the invasive margin (IM). We evaluated CD3 and CD8 cell infiltration and counted positively stained cells for each region (CT or IM) under ×400 magnification (the mean percentage of positive cells in five fields was analysed). As shown in Supplementary Fig. 7B, C, the scoring standard was 0 points for <5% CD3CT-positive cells, 1 point for 5–25% CD3CT-positive cells, 2 points for 26–50% CD3CT-positive cells and 3 points for >50% CD3CT-positive cells. The same scoring standard was used for CD3IM, CD8CT and CD8IM. Point 0 or 1 was defined as 'Low', whereas point 2 or 3 was defined as 'High'. The total TIL score IS = CD3CT + CD3IM + CD8CT + CD8IM and GC patients were stratified according to the immunoscore (TIL) reported as 0-1-2-3-4, as shown in Supplementary Fig. 7D.

We performed MLH1 (ab92312, Abcam, 1 : 250), MSH2 (ab52266, Abcam, 1 : 250), MSH6 (ab92471, Abcam, 1 : 250), PMS2 (ab110638, Abcam, 1 : 250) immunohistochemical staining and EBER (ISH-6021, ZSGB-BIO) in situ hybridization on the tissue of 444 patients[52]. The scoring criteria (Supplementary Fig. 8) were as follows: at least one mismatch repair gene-related protein was missing, interpreted as deficient mis-match repair (MMR), manifested as MSI-H; no mismatch repair gene-related protein missing was interpreted as proficient MMR, manifested as MSI-L/MSS.

Two pathologists independently scored all samples and the two doctors were unaware of the patient's clinical pathology and prognostic information. Approximately 91% of the scoring results were completely consistent. When the scores of two independent pathologists diverged, an additional pathologist reviewed the results again and chose one of the first two doctors' scores, or the three pathologists discussed the decision together.

**Multiplexed immunofluorescence staining**. Multiple immunofluorescence staining was performed to identify the expression of six indicators (NECTIN2 (95333S, CST, 1 : 100), CEACAM1 (44464S, CST, 1 : 200), HMGB1 (ab18256, Abcam, 1 : 800), SIGLEC6 (ab38581, Abcam, 1 : 100), CD44 (3570S, CST, 1 : 50), CD155 (81254S, CST, 1 : 200)), CD45 (ab10558, Abcam, 1 : 100) and panCK (ab7753, Abcam, 1 : 100) in 135 types of GC tissues. Simply put, the formalin-fixed paraffin-embedded tissue sections were cut into 4 mm-thick sections, thawed at 70 °C for 45 min, then deparaffinized and fixed the tissue with formaldehyde : methanol (1 : 10). Then, in a pH 8.0 EDTA buffer, heat-induced antigen recovery was performed at 100% power in an 800 W standard microwave until the boiling point and then 30% power was used for 15 min. The tissue sections were then cooled and washed in 0.02% tris-buffered saline-Tween 20 (TBST) with gentle stirring. After that, the sections were blocked with blocking buffer (Dako, X0909) for 10 min at room temperature and then incubated with the primary antibody at 4 °C overnight. Then, the horseradish peroxidase (HRP)-conjugated secondary antibody (PerkinElmer) was incubated at room temperature for 1 h and then the tyramide-based HRP was activated at 37 °C for 20 min. The stained signal was further amplified using Opal 540 Acetamide Signal Amplification (TSA) reagent (PerkinElmer) and incubated with TSA dilution at room temperature. Using TSA, HRP-conjugated secondary antibodies mediate the covalent binding between Pax-5 protein and different fluorophores. After this covalent reaction, perform additional antigen recovery (pH 6.0 citrate buffer) for 20 min to remove the bound antibody. Note: Repeat all steps in sequence for each primary antibody. Then, after coun-terstaining with 4′,6-diamidino-2-phenylindole (Life Technologies) at room temperature, all sections were washed five times in 0.02% TBST for 5 min, each for 2 min, and stored in a 4 °C lightproof box C until imaging.

**Literature search**. We performed a systematic literature search of PubMed before January 2019 for possible publications (Supplementary Table 2). Reports cited the references identified in this systematic review and relevant reviews were also searched to include potentially missed studies. The following terms were used in the search procedure: ('immunosuppression' or 'immune checkpoint' or 'immu-nosuppressive ligand') and ('cancer' or 'malignant tumour' or 'carcinoma' or 'tumour'). The retrieved studies were carefully examined to exclude potential duplicates or overlapping data. The titles and abstracts of articles selected from the initial search were first scanned, and then the full papers of potential eligible studies were reviewed.

**Reporting summary**. Further information on research design is available in the Nature Research Reporting Summary linked to this article.

## Data availability

The data supporting the findings in this study are available in the Article, Supplementary Information or from the corresponding author upon reasonable request.

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

## Acknowledgements

We thank members of the Key Laboratory of the Ministry of Education for Gastro-intestinal Cancer for helpful comments and suggestions.

## Author contributions

J.B.W., P.L., and X.L.L. conceived the study and instructed the manuscript. Y.H.Y., C.M.H., and C.H.Z. helped critically revise the manuscript for important intellectual content. Q.L.Z., L.C.L., and N.Z.L. analysed the data and drafted the manuscript. Y.B.M., Y.J.Z., J.W.X., J.X.L., J.L., L.L.C., and M.L. helped collect data and design the study.

## Conflict of interest

The authors declare no competing interest.
