## [Peer Review File · Nature Communications]

Reviewers' comments:

Reviewer #1 (Remarks to the Author):

The authors present an interesting analysis starting with initial identification of 20 immune checkpoint molecules with expression measured by IHC in TMAs, ultimately settling on 6 markers in their scoring system for immunosuppression in resected gastric cancer (ISS-GC). Strength of this report includes the large number of cases in both discovery, training, and validation sets in which the authors' scoring system in addition to a previously established clinical scoring system (PNI) carries prognostic significance in their patient populations and possibly predict for benefit from adjuvant chemotherapy.

A major perceived weakness of this analysis is the lack of determination of the genomic classification of their resected gastric cancer cases, specifically in relation to the gastric cancer TCGA classification [1]. On a molecular basis the MSI-H and EBV-associated subtypes of gastric cancer are typically associated with upregulation of multiple immune checkpoints [2], and as such it would be important to discern that the ISS-GC groups the authors observe are not exclusive to TCGA molecular subtypes. Furthermore, other groups have demonstrated differing prognosis of resected gastric cancers when categorized by TCGA molecular subtyping, and also differential benefit of adjuvant chemotherapy in post-hoc analyses [3]. More recently, identification of patients that are MSI-H even with traditional DNA-PCR techniques without need for high throughput sequencing have demonstrated the more favorable prognosis and lack of benefit from adjuvant/perioperative chemotherapy in non-metastatic gastric cancer [4]. I would suggest as a major revision the authors determine MSI and EBV status of their cases and re-analyze the prognostic and adjuvant-chemotherapy predictive value of their ISS-GC score. The novelty and scientific relevance would be increased if their scoring system can provide additional immune microenvironment characterization among MSS, EBV-negative gastric cancers.

References

1. The Cancer Genome Atlas Research, N., Comprehensive molecular characterization of gastric adenocarcinoma. *Nature*, 2014. 513(7517): p. 202-209.
2. Panda, A., et al., Immune Activation and Benefit From Avelumab in EBV-Positive Gastric Cancer. *J Natl Cancer Inst*, 2018. 110(3): p. 316-320.
3. Sohn, B.H., et al., Clinical Significance of Four Molecular Subtypes of Gastric Cancer Identified by The Cancer Genome Atlas Project. *Clinical Cancer Research*, 2017. 23(15): p. 4441-4449.
4. Pietrantonio, F., et al., Individual Patient Data Meta-Analysis of the Value of Microsatellite Instability As a Biomarker in Gastric Cancer. *J Clin Oncol*, 2019. 37(35): p. 3392-3400.

Peer review report by:

Joseph Chao, MD

Department of Medical Oncology and Therapeutics Research

City of Hope Comprehensive Cancer Center

Duarte, CA, USA

Reviewer #2 (Remarks to the Author):

The authors profiled the expression levels of 20 putative checkpoint markers with IHC for ~800 gastric cancer patients. They selected 6 markers as prognostic predictors using the training cohort, and performed Cox regression using the test and validation cohorts to show that the immune score is an independent predictor of patient outcome. Overall, the purpose of this work to identify novel prognostic markers for GC is well justified, but the approach suffers serious problems that need to be addressed:

1. The authors used 6 markers to define ISSgc. In their formula, the coefficients are both positive and negative. For example, CEACAM1 is positive, and NECTIN2 negative. I understand that the

Cox model provided these estimates, but can the authors provide a rational explanation? Intuitively, the coefficients should be related to the hazard ratios, but looking at Figure S3, it seems that higher expression of all the markers is associated with worse outcome.

2. What are the source of origins expressing these markers? Since this study focuses on the immune checkpoint pathways, it is critical for the authors to distinguish if the selected markers were expressed mainly in cancer or in immune cells. Further, it will be helpful to clarify my next question (see below).

3. There seem to be a trivial explanation to the observed clinical relevance of ISSGc: is this simply caused by higher or lower infiltration of certain immune cells, or the overall infiltration of all the CD45+ cells? It will be helpful to evaluate this possibility if the authors can provide an overall immune infiltration and/or lymphocyte infiltration score, and investigate if the protein level of any of the 6 markers is significantly correlated with infiltration. This is particularly interesting for this study because the patients were not treated with any immunotherapy. Therefore, either ISSGc or PNI was not induced by treatment, but reflected the general competence of the patient immune system, which will likely affect immune infiltration as well.

4. Is ISSGc score evenly distributed across different TNM stages? Tumors with more advanced stages are expected to have higher immune suppression. Also, if this correlation is observed, how does that influence the Cox regression models in this work?

5. The statistical interaction between ISSGc and chemotherapy is intriguing, but the analysis needs to be more rigorous. First, the authors need to show the exact P values on the plot, instead of >0.05 or <0.05 . Second, why authors chose to combine stages II and III in Figure 3B, but split them in Figure 3DEGH? It seems that the survival difference between chemo treated patients in the high ISSGc patients is quite substantial, but failed to reach significance due to smaller sample size. If increasing sample cohort brings back significance, the authors' conclusion could be misleading.

Reviewer #3 (Remarks to the Author):

I am mostly qualified to comment on the statistical aspects of the manuscript. The manuscript is well written, but it would benefit from involving a biostatistician and a more thorough analysis. My comments are below:

1. Details of the statistical analysis are missing (e.g., how was the penalization parameter be selected for the LASSO model). This leads to lack of reproducibility. Providing code even if data is not publicly available would enable readers to figure out the exact details of the statistical analysis.
2. How was missing data handled in this analysis?
3. Electronic health records suffer from many potential biases (e.g., survival bias, missing data, measurement error etc.). Are those serious concerns? For example, does inclusion criteria c) lead to survival bias? Where all variables recorded in an identical manner between the different institutions?
4. "All surviving patients were followed for more than five years". Does that mean you ignored patients who were not included in the health records for at least 5 years? If yes, that could lead to bias.
5. "Time to survival was defined as the time between the date of surgery and the date of death or April 2019." I don't understand the role of April 2019 in defining the time to survival. Please clarify
6. The use of stepward regression is considered questionable in the statistical literature. It seems like the number of features is small enough to warrant including all of them. I would recommend not using all variables without variable selection.
7. Some of the figures, such as Figure S.9, in the supplement are hard to read.
8. How was the analysis that lead to the claim "ISSGC + TNM staging in the internal cohort was

better than TNM staging alone by the C-index" performed?

9. As the authors only consider 7 covariates the use of LASSO seems not necessary. As the authors correctly point out, LASSO is beneficial for large number of covariates. Did the authors explore interactions or nonlinear transformations? In that case, the use of LASSE becomes easier to justify.

10. How were censored observations dealt with when the C-index was calculated?

Reviewer #1 (Remarks to the Author):

1. The authors present an interesting analysis starting with initial identification of 20 immune checkpoint molecules with expression measured by IHC in TMAs, ultimately settling on 6 markers in their scoring system for immunosuppression in resected gastric cancer (ISS-GC). Strength of this report includes the large number of cases in both discovery, training, and validation sets in which the authors' scoring system in addition to a previously established clinical scoring system (PNI) carries prognostic significance in their patient populations and possibly predict for benefit from adjuvant chemotherapy.

Response: Thanks very much for the time and efforts of the reviewer. We sincerely appreciate that the reviewer commented our work as "interesting analysis".

0. A major perceived weakness of this analysis is the lack of determination of the genomic classification of their resected gastric cancer cases, specifically in relation to the gastric cancer TCGA classification [1]. On a molecular basis the MSI-H and EBV-associated subtypes of gastric cancer are typically associated with upregulation of multiple immune

checkpoints [2], and as such it would be important to discern that the ISS-GC groups the authors observe are not exclusive to TCGA molecular subtypes. Furthermore, other groups have demonstrated differing prognosis of resected gastric cancers when categorized by TCGA molecular subtyping, and also differential benefit of adjuvant chemotherapy in post-hoc analyses [3]. More recently, identification of patients that are MSI-H even with traditional DNA-PCR techniques without need for high throughput sequencing have demonstrated the more favorable prognosis and lack of benefit from adjuvant/perioperative chemotherapy in non-metastatic gastric cancer [4]. I would suggest as a major revision the authors determine MSI and EBV status of their cases and re-analyze the prognostic and adjuvant-chemotherapy predictive value of their ISS-GC score. The novelty and scientific relevance would be increased if their scoring system can provide additional immune microenvironment characterization among MSS, EBV-negative gastric cancers.

References

- [1] The Cancer Genome Atlas Research, N., Comprehensive molecular characterization of gastric adenocarcinoma. *Nature*, 2014, 513(7517): 202-209.
- [2] Panda A, et al. Immune Activation and Benefit From Avelumab in EBV-Positive Gastric Cancer. *J Natl Cancer Inst*, 2018, 110(3): 316-320.
- [3] Sohn BH, et al. Clinical Significance of Four Molecular Subtypes of Gastric Cancer Identified by The Cancer Genome Atlas Project. *Clinical Cancer Research*, 2017, 23(15): 4441-4449.
- [4] Pietrantonio F, et al. Individual Patient Data Meta-Analysis of the Value of Microsatellite Instability As a Biomarker in Gastric Cancer. *J Clin Oncol*, 2019, 37(35): 3392-3400.

Peer review report by:

Joseph Chao, MD

Department of Medical Oncology and Therapeutics Research

City of Hope Comprehensive Cancer Center

Duarte, CA, USA

Response: Thank you very much for the extremely helpful comments. Following the reviewer's suggestion, we have further determined the MSI and EBV status in our own cohort. The accumulation of mismatches during DNA replication due to loss of DNA mismatch repair (MMR) gene expression is the cause of MSI^{1,2}. At present, immunohistochemistry (IHC) is commonly used to detect the expression of mismatch repair (MMR) gene-related proteins (MLH1, PMS2, MSH2 and MSH6) to determine the MSI status of gastric cancer patients³. If IHC showed that at least one mismatch repair gene-related protein was missing, it was interpreted as deficient MMR (dMMR), manifested as MSI-H; no mismatch repair gene-related protein was identified as proficient MMR (pMMR), manifested as MSI-L/MSS. At the same time, EBV-RNA (EBER) detection by situ hybridization is also commonly used in the clinic to diagnose the presence of EBV in gastric cancer tissues. Therefore, we performed MLH1, MSH2, MSH6, PMS2 immunohistochemical staining and EBER in situ hybridization on 444 patient cancer tissues to determine their MSI and EBV status in Figure 1 (also as Supplemental Figure S8 in the revised manuscript). Kaplan-Meier analysis and stratification analysis were performed to evaluate the prognostic value of ISS_{GC} after stratification by MSI or EBV-associated subtypes. As shown in Figures 2A - 2D (also as Figures 5A - 5D in the revised manuscript), after stratification by MSI subtypes or EBV-associated subtypes, ISS_{GC} was still significantly correlated with the prognosis of all subtypes: a higher ISS_{GC} indicated a worse prognosis independent of MSI and EBV status, indicating that the prognostic value of ISS_{GC} did not need to consider MSI and EBV status ($p < 0.001$). After stratification by MSI subtype and receipt of chemotherapy, Kaplan-Meier and stratification analysis showed that patients in the low- ISS_{GC} group either with MSI-H or MSI-L/MSS all had a better prognosis when they received chemotherapy, while the high- ISS_{GC} group had no additional significant benefit regarding overall survival with chemotherapy in both MSI-H and MSI-L groups in Figures 2E - 2H below (also as Figures 5E - 5H in the revised manuscript). Similarly, after stratification by EBV-associated subtypes and receipt of chemotherapy, we also found that the low- ISS_{GC} group of EBV-negative patients could benefit from adjuvant chemotherapy and had a better prognosis than those without chemotherapy, while the high- ISS_{GC} group of EBV-negative patients could not benefit from adjuvant chemotherapy and had no

significant difference in prognosis regardless of whether they received chemotherapy, as shown in Figures 2I - 2J (also as Figures 5I – 5J in the revised manuscript). However, we could not analyse the effects of chemotherapy on the prognosis of EBV-positive gastric cancer patients due to the small number of patients. We also found similar results in the external validation cohort, as shown in Figure 3 below (also as Figure 8 in the revised manuscript).

In summary, the _{ISSGC} scoring system was able to distinguish patients with poor prognosis and screen patients who might benefit from adjuvant chemotherapy independent of their MSI or EBV status and could further provide additional immune microenvironment characterization among MSI-L/MSS and EBV-negative gastric cancer patients.

We have added these methods to the ‘Immunohistochemistry’ section of the ‘Supplemental Materials and Methods’ in the revised manuscript. Additionally, we have added these results to the 4th and 6th points of the ‘Results’ section in the revised manuscript.

Figure 1. Immunohistochemical scoring criteria for MSI and in situ hybridization scoring criteria for EBV status under x40 magnification.

Figure 2. Prognostic value of ISS_{GC} score on gastric cancer patients and receipt of adjuvant chemotherapy stratified by their MSI or EBV status. (A-B) Overall survival based on MSI status in GC patients after ISS_{GC} stratification is presented using Kaplan-Meier curves. (C-D) Overall survival based on EBV status in GC patients after ISS_{GC} stratification is presented using Kaplan-Meier curves. (E-H) Overall survival analysis of chemotherapy based on MSI status in GC patients after ISS_{GC} stratification is presented using Kaplan-Meier curves. (I-J) Overall survival analysis of chemotherapy in EBV negative GC patients after ISS_{GC} stratification is presented using Kaplan-Meier curves.

Figure 3. External validation. Prognostic value of ISSGC score on gastric cancer patients and receipt of adjuvant chemotherapy stratified by their MSI or EBV status. (A-B) Overall survival based on MSI status in GC patients after ISSGC stratification is presented using Kaplan-Meier curves. (C-D) Overall survival based on EBV status in GC patients after ISSGC stratification is presented using Kaplan-Meier curves. (E-H) Overall survival analysis of chemotherapy based on MSI status in GC patients after ISSGC stratification is presented using Kaplan-Meier curves. (I-J) Overall survival analysis of chemotherapy in EBV negative GC patients after ISSGC stratification is presented using Kaplan-Meier curves.

References

- 1 Cristescu R, et al. Molecular analysis of gastric cancer identifies subtypes associated with distinct clinical outcomes. *Nat. Med.* **21**, 449-456, (2015).

- 2 Romain C, et al. Immune Checkpoint Inhibition in Colorectal Cancer: Microsatellite Instability and Beyond, Targeted Oncology **15**, 11-24 (2019).
- 3 Emily M Lin, Joseph Chao, et al. Advances in immuno-oncology biomarkers for gastroesophageal cancer: programmed death ligand 1, microsatellite instability, and beyond, World J Gastroenterol **24**, 2686-2697 (2018).
- 4 Esmeralda CM, et al. Is There a Role for Programmed Death Ligand-1 Testing and Immunotherapy in Colorectal Cancer With Microsatellite Instability? , Arch Pathol Lab Med **12** (2018).

Reviewer #2 (Remarks to the Author):

The authors profiled the expression levels of 20 putative checkpoint markers with IHC for ~800 gastric cancer patients. They selected 6 markers as prognostic predictors using the training cohort, and performed Cox regression using the test and validation cohorts to show that the immune score is an independent predictor of patient outcome. Overall, the purpose of this work to identify novel prognostic markers for GC is well justified, but the approach suffers serious problems that need to be addressed:

1. The authors used 6 markers to define ISS_{GC} . In their formula, the coefficients are both positive and negative. For example, CEACAM1 is positive, and NECTIN2 negative. I understand that the Cox model provided these estimates, but can the authors provide a rational explanation? Intuitively, the coefficients should be related to the hazard ratios, but looking at Figure S3, it seems that higher expression of all the markers is associated with worse outcome.

Response: Thank you very much. In this study, we used KM analysis to screen out 6 indicators that were highly expressed in tumours and showed survival significance. Mechanistically, CD44 reduced the sensitivity of tumour cells to cytotoxic T cells (CTLs) by downregulating the Fas-FasL pathway, causing tumours to escape CTL killing; CEACAM1 affected the immune tolerance of T cells through the TIM-3 signalling pathway and suppressed the immune response of T cells to cancer cells; both SIGLEC6 and HMGB1 played an immunosuppressive function by regulating the activity of mast cells, and they

might check each other when they played the same role; CD155 and NECTIN2 were both members of the Nectin-like molecule family, which affected the activity of CTLs, thereby inhibiting antitumour immunity, and the function of the two might be mutually constrained. Therefore, due to the co-stimulation or mutual inhibition between immunosuppressive molecules, there would be correlations among multiple indicators. When the variables are correlated, the traditional Cox model is not applicable, so we choose the LASSO model to establish the ISS scoring system. The LASSO model compresses high-dimensional data by compressing the coefficients of some meaningless variables to 0. It can eliminate offsets caused by correlations between variables, resulting in a more stable scientific model ¹⁻³. When the coefficients of some variables are small and tend to 0, the variable's contribution to the overall formula is small. We also conducted a statistical correlation analysis to confirm the correlation between the 7 indicators in Table 1 below (also as Supplemental Table S3 in the revised manuscript). Among them, the correlation between NECTIN2 and CD155 was significant, as was that between HMGB1 and SIGLEC6. Those correlations led to the fact that although NECTIN2 and HMGB1 were highly expressed in cancer and were associated with poor prognosis, their role in the ISS formula is constrained by other indicators, so the coefficients of two in the ISS formula were negative. Moreover, the coefficients of NECTIN2 and HMGB1 were -0.0189 and -0.0053, respectively, which were both close to 0. Therefore, the contribution of NECTIN2 and HMGB1 to the ISS formula was smaller than that of the other four indicators, although their coefficients are negative.

We have added the Spearman correlation analysis results to the 1st point of the 'Results' section in the revised manuscript.

References

- 1 Tibshirani R, et al. The lasso method for variable selection in the Cox model, *Stat Med.* **16**, 385-395 (1997).
- 2 Tibshirani R, et al. Regression shrinkage and selection via the lasso: a retrospective. *J Roy Stat Soc Series B Stat Methodol* **73**, 273-282 (2011).
- 3 Zou H, et al. The adaptive lasso and its oracle properties, *Journal of the American Statistical Association* **101**, 1418-1429 (2006).

Table 1. Spearman correlation analysis of training set

Correlation coefficient	SIGLEC6	CD44	CD155	HMGB1	NECTIN2	CEACAM1	ADENOSINE
SIGLEC6	1	0.061	0.192**	0.297**	0.202**	0.160**	0.328**
CD44	0.061	1	0.052	0.094*	0.045	-0.033	0.133**
CD155	0.192**	0.052	1	0.205**	0.464**	0.342**	0.135**
HMGB1	0.297**	0.094*	0.205**	1	0.158*	0.087	0.179**
NECTIN2	0.202**	0.045	0.464**	0.158*	1	0.384**	0.105*
CEACAM1	0.160**	-0.033	0.342**	0.087	0.384**	1	0.008
ADENOSINE	0.328**	0.133**	0.135**	0.179**	0.105*	0.008	1

** $p < 0.01$ * $p < 0.05$

2. What are the source of origins expressing these markers? Since this study focuses on the immune checkpoint pathways, it is critical for the authors to distinguish if the selected markers were expressed mainly in cancer or in immune cells. Further, it will be helpful to clarify my next question (see below).

Response: Thank you very much. Following the reviewer's suggestion, we performed multi-colour immunofluorescence staining analysis on the 6 immune indicators together with CKpan and CD45 to determine the origin of these 6 indicators. As shown in Figure 4 below (also as Figure 2 in the revised manuscript), the results indicated that all 6 indicators were colocalized with CKpan; therefore, they should mostly originate from gastric cancer cells.

We have added these methods and results to the 'Multiplexed immunofluorescence staining' part of the 'Supplemental Materials and Methods' section and the 1st point of the 'Results' section in the revised manuscript.

Figure 4. Multi-colour immunofluorescence staining of anti-CKpan and CD45 antibodies with CD155, NECTIN2, CEACAM1, HMGB1, SIGLEC6 or CD44 antibodies.

3. There seem to be a trivial explanation to the observed clinical relevance of ISSgc: is this simply caused by higher or lower infiltration of certain immune cells, or the overall infiltration of all the CD45+ cells? It will be helpful to evaluate this possibility if the authors can provide an overall immune infiltration and/or lymphocyte infiltration score, and investigate if the protein level of any of the 6 markers is significantly correlated with infiltration. This is particularly interesting for this study because the patients were not treated with any immunotherapy. Therefore, either ISSgc or PNI was not induced by treatment, but reflected the general competence of the patient immune system, which will likely affect immune infiltration as well.

Response: Thank you very much. Following the reviewer's excellent suggestion, we performed CD45, CD3 and CD8 immunohistochemical staining on the tumour tissue of 444 patients to evaluate the overall immune infiltration and tumour infiltrating lymphocyte (TIL) infiltration. The expression of CD45 reflects overall immune infiltration. We evaluated CD45 cell infiltration and counted positively stained cells in each tumour region under x400 magnification (the mean percentage of positive cells in 5 fields was analysed), and the

scoring standard was 0 points for <5% CD45-positive cells, 1 point for 5%-25% CD45-positive cells, 2 points for 26%-50% CD45-positive cells, and 3 points for >50% CD45-positive cells, as shown in Figure 5A below (also as Supplemental Figure S7A in the revised manuscript). Until now, there has been no current consensus or international guidelines on the morphologic evaluation of TILs in GC, so we adopted and modified the GALON scoring, which used CD3 and CD8 as markers for reflecting the condition of TILs¹⁻⁵ in colorectal cancer. The tumour area was divided into the centre of the tumour (CT) and the invasive margin (IM). We evaluated CD3 and CD8 cell infiltration and counted positively stained cells for each region (CT or IM) under x400 magnification (the mean percentage of positive cells in 5 fields was analysed). As shown in Figures 5B and 5C (also as Supplemental Figures S7B and 7C in the revised manuscript), the scoring standard was 0 points for <5% CD3_{CT}-positive cells, 1 point for 5%-25% CD3_{CT}-positive cells, 2 points for 26%-50% CD3_{CT}-positive cells, and 3 points for >50% CD3_{CT}-positive cells. The same scoring standard was used for CD3_{IM}, CD8_{CT} and CD8_{IM}. Point 0 or 1 was defined as 'Low', while point 2 or 3 was defined as 'High'. Immunoscore (IS: the total TIL score) = CD3_{CT} + CD3_{IM} + CD8_{CT} + CD8_{IM} and GC patients were stratified as I0-1-2-3-4, as shown in Figure 5D below (also as Supplemental Figure S7D in the revised manuscript). Spearman correlation analysis showed that there was no correlation between CD45 overall infiltration or TIL infiltration and 6 indicators, as shown in Table 2 below (also as Supplemental Table S4 in the revised manuscript), which indicated that the six indicators were not related to the overall immune infiltration or the tumour infiltrating lymphocytes. Therefore, these 6 indicators might mediate the immunosuppression of CTL activation rather than mediate immune cell infiltration. We have supplemented this conclusion in the results and discussion section, which may be the direction of our future research on tumour immunosuppression.

We have added these methods and results to the 'Immunohistochemistry' section of the 'Supplemental Materials and Methods' section and the 1st point of the 'Results' section in the revised manuscript.

Figure 5. (A) Immunohistochemical scoring criteria for CD45 (B-C) Immunohistochemical scoring criteria for CD3_{CT}, CD3_{IM}, CD8_{CT}^{and} CD8_{IM} (D) Immunoscore definition for TIL

Table 2. Spearman correlation analysis of training set(n=444)

Variable	TIL		CD45	
	r	P	r	P
SIGLEC6	-0.008	0.867	0.004	0.938
CD44	0.073	0.122	0.060	0.208
CD155	0.061	0.200	0.046	0.336
HMGB1	-0.205	0.593	0.057	0.232
NECTIN2	0.095	0.057	0.086	0.071
CEACAM1	0.059	0.214	0.000	0.993

References

- 1 Jerome Galon, et al. Type, Density, and Location of Immune Cells Within Human Colorectal Tumors Predict Clinical Outcome, *SCIENCE* **313**, 1960-1964 (2006).
- 2 Jerome Galon, et al. Towards the introduction of the 'Immunoscore' in the classification of malignant tumours, *Journal of Pathology* **232**, 199-209 (2014).
- 3 Jerome Galon, et al. Cancer classification using the Immunoscore: a worldwide task force, *Journal of Translational Medicine* **10**, 205-214 (2012).
- 4 Maria-Gabriela Anitei, et al. Prognostic and Predictive Values of the Immunoscore in Patients with Rectal Cancer, *Clin Cancer Res* **20**, 1891-1899 (2014).
- 5 Mouna Trabelsi, et al. An Immunoscore System Based On CD3+ And CD8+ Infiltrating Lymphocytes Densities To Predict The Outcome Of Patients With Colorectal Adenocarcinoma, *Oncotargets and Therapy* **12**, 8663-8673 (2019).

4. Is ISS_{GC} score evenly distributed across different TNM stages? Tumors with more advanced stages are expected to have higher immune suppression. Also, if this correlation is observed, how does that influence the Cox regression models in this work?

Response: Thank you very much. As you can see in Table 3 below, we found that there were more patients with high ISS_{GC} in advanced gastric cancer (stage II or III patients) than in early gastric cancer (stage I patients). TNM stage was significantly correlated with ISS_{GC} score ($r = 0.155$, $P = 0.001$) using Spearman correlation analysis. To eliminate the offset effect caused by the correlation, we used Cox multivariate analysis to correct it. TNM stage and ISS_{GC} score were still independent factors for patient prognosis. In addition, the results of stratified analysis also showed that stratified by different TNM stages, the impact of the ISS_{GC} score on the prognosis was significantly different, showing that the value of the ISS_{GC} score on the prognosis was not evenly disturbed across different TNM stages.

Table 3. Cross list of ISS and TNM stage(n=444)

TNM stage	ISS		Total
	ISS≤1.83	ISS >1.83	
I	59(93.7%)	4(6.3%)	63

II	110(79.7%)	28(20.3%)	138
III	187(77.0%)	56(23.0%)	243
Total	356(80.2%)	88(19.8%)	444

5. The statistical interaction between ISS_{GC} and chemotherapy is intriguing, but the analysis needs to be more rigorous. First, the authors need to show the exact P values on the plot, instead of >0.05 or <0.05 . Second, why authors chose to combine stages II and III in Figure 3B, but split them in Figure 3DEGH? It seems that the survival difference between chemo treated patients in the high ISS_{GC} patients is quite substantial, but failed to reach significance due to smaller sample size. If increasing sample cohort brings back significance, the authors' conclusion could be misleading.

Response: Thank you very much. We have provided the exact P value of all the survival curves in the revised manuscript following the reviewer's suggestion and have separately analysed the patients of stages II and III and added the survival curve of stages II and III

below Figure 6 (also as Figures 4B – 4C in the revised manuscript).

Figure 6. Overall survival based on TNM II and III stage in GC patients after ISS_{GC} stratification

The number of patients with $ISS_{GC} > 1.83$ in the internal cohort was relatively small, which may affect the accuracy of the results, and it might be one of the deficiencies of this article. We have added this shortcoming to the 'Discussion' section of the revised manuscript. To expand the sample size, we combined patients from the internal centre with external centres. A total of 169 patients with $ISS_{GC} > 1.83$ included 77 patients with postoperative

adjuvant chemotherapy and 92 patients who did not receive postoperative adjuvant chemotherapy. The stratified analysis after expanding samples showed no significant

difference in the prognosis of patients with or without chemotherapy when $ISS_{GC} > 1.83$ (Figure 7). This result is consistent with our previous result.

Figure 7. Overall survival analysis of chemotherapy in GC patients with $ISS_{GC} > 1.83$

Reviewer #3 (Remarks to the Author):

I am mostly qualified to comment on the statistical aspects of the manuscript. The manuscript is well written, but it would benefit from involving a biostatistician and a more thorough analysis. My comments are below:

1. Details of the statistical analysis are missing (e.g., how was the penalization parameter be selected for the LASSO model). This leads to lack of reproducibility. Providing code even if data is not publicly available would enable readers to figure out the exact details of the statistical analysis.

Response: Thank you very much. Following the reviewer's very helpful suggestion, we have added the detailed statistical methods used in this study in the Materials and Methods section: We used the t test or Mann-Whitney U test to analyse continuous variables and the χ^2 test or Fisher's exact test to compare the categorical variables of clinical features. We used the Spearman test or Kruskal-Wallis test for correlation analysis, and the prognosis analysis was analysed by the Kaplan-Meier method. Seven meaningful indicators (HMGB1, NECTIN2, SIGLEC6, CEACAM1, ADENOSINE, CD44 and CD155)

were selected from 20 indicators. A stepwise regression method was used in the relevant clinical pathology variable analysis. The clinical pathology data were first subjected to single-factor Cox analysis, and then the selected meaningful variables were included in the multivariate regression analysis. The $ISSGC$ classifier was built by using the LASSO regression (kernel Cox) model. The data were analysed using EmpowerR software (version 2.0), and the corresponding code was uploaded (Annex 1). The C-index software package and AIC software package of R software (version 4.0.0) were used to evaluate and compare the model's ability to predict survival. We have modified and supplemented the 'Statistical analysis' part of the 'Materials and Methods' section in the revised manuscript.

2. How was missing data handled in this analysis?

Response: In this study, the data of 444 patients in the modelling group of this centre were relatively complete, with no missing data (Table 1 in the revised manuscript). The adjuvant chemotherapy, survival, and PNI index of 226 patients in the external centre validation group were complete. We also updated and reanalysed the TNM stage, age, gender, tumour size and other data of patients in the external validation group (Supplemental Table S1 in the revised manuscript). For patients in the external validation group who lost continuous variable BMI data, the average value was substituted in this study, and it did not affect the validation of the $ISSGC$ model in this study. We have modified and supplemented the 'Clinicopathological data' part of the 'Materials and Methods' section in the revised manuscript.

3. Electronic health records suffer from many potential biases (e.g., survival bias, missing data, measurement error etc.). Are those serious concerns? For example, does inclusion criteria c) lead to survival bias? Where all variables recorded in an identical manner between the different institutions?

Response: Thank you very much. The three centres were individually staffed to collect, organize, enter, and follow up the data in accordance with the provisions of Statute ¹ and

performed unified maintenance on the database on a regular basis. This study included the clinical and pathological data of patients in the three centres. The difference in recording personnel and medical equipment would produce data measurement errors, resulting in potential offset effects, which might also be a deficiency of this article. However, the expression and score of each immunosuppression index were evaluated by the same two pathologists according to a unified scoring standard to exclude the result bias caused by measurement errors. At the same time, the key clinical indicators of the three centres, such as the patient's tumour pathology type, TNM stage, postoperative chemotherapy status, and survival status, were all checked by the dedicated staff to minimize deviation. We have modified and supplemented the 'Clinicopathological data' part of the 'Materials and Methods' section in the revised manuscript.

References

- 1 Japanese Gastric Cancer Association, Japanese gastric cancer treatment guidelines 2010 (ver. 3), *Gastric Cancer* **14**, 113-123 (2011).

4. "All surviving patients were followed for more than five years". Does that mean you ignored patients who were not included in the health records for at least 5 years? If yes, that could lead to bias.

Response: Thank you very much. This study included a total of 794 gastric cancer patients from the Union Hospital of Fujian Medical University, the Affiliated Hospital of Qinghai University, and the First Affiliated Hospital of University of Science and Technology of China from January 2010 to April 2014. The last follow-up time was April 2019. The survival time of deceased patients is defined as the time from surgery to the time of death, and the survival time of surviving patients is defined as the time from surgery to the time of the last follow-up. Our follow-up method was conducted by clinicians of the three centres in accordance with the unified standards of the Japanese Statute ¹. This study introduced all patients to statistical analysis, including patients lost to follow-up. The overall loss ratio of follow-up was 4.4%; therefore, it had little impact on this study.

References

1 Japanese Gastric Cancer Association, Japanese gastric cancer treatment guidelines 2010 (ver. 3), *Gastric Cancer* **14**, 113-123 (2011).

5. "Time to survival was defined as the time between the date of surgery and the date of death or April 2019." I don't understand the role of April 2019 in defining the time to survival. Please clarify.

Response: Thank you very much. In this study, we defined overall survival as the time from surgery to the time of the last follow-up (April 2019), the time of death, or the database deadline (time lost to follow-up). The survival time of deceased patients was defined as the time from surgery to death, and the survival time of surviving patients was defined as the time from surgery to the time of the last follow-up. We have added these definitions to the 'Follow up' part of the 'Materials and Methods' section in the revised manuscript.

6. The use of stepward regression is considered questionable in the statistical literature. It seems like the number of features is small enough to warrant including all of them. I would recommend not using all variables without variable selection.

Response: Thank you very much. Stepwise regression is used to select variables that are meaningful for univariate regression and then incorporate them into the analysis of multifactor regression. Its role is to eliminate variables that have no significant effect on the results ¹. In this study, we first performed a univariate Cox analysis on the clinical pathological data and then included the selected meaningful variables in the multivariate regression analysis to eliminate variables that did not significantly affect the results. The inclusion criterion for stepwise regression analysis is $P_e = 0.05$, and the elimination criterion is $P_r = 0.1$. We have supplemented the 'Statistical analysis' part of the 'Materials and Methods' section with a detailed description of the stepwise regression method in the revised manuscript.

References

1 Grambsch PM, Therneau TM, et al. Proportional Hazards Tests and Diagnostics Based on Weighted Residuals, *Biometrika* **81**, 515-526 (1994).

7. Some of the figures, such as Figure S.9, in the supplement are hard to read.

Response: Thank you very much. The number of patients was so large that the heat map was too long, and the legend was unclear. We have re-uploaded Figure S11 in the revised manuscript with clear legends. The abscissa represents the patient number of 444 patients in the internal centre, and the ordinate represents the expression of CD155, NECTIN2, CEACAM1, HMGB1, SIGLEC6, ADENOSINE, CD44, TIL and CD45.

8. How was the analysis that led to the claim "ISSGC + TNM staging in the internal cohort was better than TNM staging alone by the C-index" performed?

Response: Thank you very much. We found that there were studies that only used the C-index or C-index + AIC/BIC to compare the prognosis evaluation of the two models 1-7. Therefore, in this study, we used the C-index to compare the evaluation of the prognosis of the ISS + TNM + MSI model with that of the TNM + MSI model. In addition, we used the AIC index to evaluate two models and drew the same conclusion (internal verification: AIC (ISS + TNM + MSI) = 2254.93 < AIC (TNM + MSI) = 2300.13; external verification: AIC (ISS + TNM + MSI) = 855.23 < AIC (TNM + MSI) = 861.09). We have added these results to the 4th and 6th points of the 'Results' section in the revised manuscript.

References

- 1 Han D, Suh YS, Kong SH, et al. Nomogram Predicting Long-Term Survival After D2 Gastrectomy for Gastric Cancer, *Journal of Clinical Oncology* **30**, 3834-3840 (2012).
- 2 Zhou R, Zhang J, Zeng D, et al. Immune cell infiltration as a biomarker for the diagnosis and prognosis of stage I-III colon cancer, *Cancer Immunology, Immunotherapy* **68**, 433-442 (2019).
- 3 Qu A, Yang YM, et al. Development of a preoperative prediction nomogram for lymph node metastasis in colorectal cancer based on a novel serum miRNA signature and CT scans, *EBioMedicine* **52**, 1-9 (2018).
- 4 Hou X, Wang D, et al. Development and validation of a prognostic nomogram for HIV/AIDS patients who underwent antiretroviral therapy: Data from a Chinapopulation-based cohort, *EBioMedicine* **31**, 1-11 (2019).

- 5 Fang C, Wang W, et al. Nomogram individually predicts the overall survival of patients with gastroenteropancreatic neuroendocrine neoplasms, *British Journal of Cancer*, **315**, 1-7 (2017).
- 6 Wen J, Wang G, et al. Prognostic Value of a Four-miRNA Signature in Patients with Lymph Node Positive Locoregional Esophageal Squamous Cell Carcinoma Undergoing Complete Surgical Resection, *Annals of Surgery*, doi:10.1097/SLA.0000000000003369 (2019).
- 7 Ulrich N, Matthias M, et al. Prediction of Prognosis Is Not Improved by the Seventh and Latest Edition of the TNM Classification for Colorectal Cancer in a Single-Center Collective, *Annals of Surgery*, **254**, 794-801 (2011).

9. As the authors only consider 7 covariates the use of LASSO seems not necessary. As the authors correctly point out, LASSO is beneficial for large number of covariates. Did the authors explore interactions or nonlinear transformations? In that case, the use of LASSO becomes easier to justify.

Response: Thank you very much. Although only 7 covariates were included in this study for LASSO analysis, Spearman correlation analysis found correlations between indicators in Table 4 (also as Supplemental Table S3 in the revised manuscript). When the variables are correlated, the traditional Cox model is not applicable, so we chose the LASSO model to establish the ISS scoring system. The LASSO model compresses high-dimensional data by compressing the coefficients of some meaningless variables to 0. It can eliminate offsets caused by correlations between variables, resulting in a more stable scientific model ¹⁻³. We have added the advantages of LASSO and the Spearman correlation analysis results to the 'Statistical analysis' part of the 'Materials and Methods' section and the 1st point of the 'Results' section in the revised manuscript, respectively.

Table 4. Spearman correlation analysis of training set (n=444)

Correlation coefficient		SIGLEC6	CD44	CD155	HMGB1	NECTIN2	CEACAM1	ADENOSINE
SIGLEC6	1	0.061	0.192**	0.297**	0.202**	0.160**	0.328**	

CD44	0.061	1	0.052	0.094*	0.045	-0.033	0.133**
CD155	0.192**	0.052	1	0.205**	0.464**	0.342**	0.135**
HMGB1	0.297**	0.094*	0.205**	1	0.158*	0.087	0.179**
NECTIN2	0.202**	0.045	0.464**	0.158*	1	0.384**	0.105*
CEACAM1	0.160**	-0.033	0.342**	0.087	0.384**	1	0.008
ADENOSINE	0.328**	0.133**	0.135**	0.179**	0.105*	0.008	1

** $p < 0.01$ * $p < 0.05$

References

- 1 Tibshirani R, et al. The lasso method for variable selection in the Cox model, *Stat Med.* **16**, 385-395 (1997).
- 2 Tibshirani R, et al. Regression shrinkage and selection via the lasso: a retrospective. *J Roy Stat Soc Series B Stat Methodol*, **73**, 273-282 (2011).
- 3 Zou H, et al. The adaptive lasso and its oracle properties, *Journal of the American Statistical Association*, **101**, 1418-1429 (2006).

10. How were censored observations dealt with when the C-index was calculated?

Response: Thank you very much. The censoring when calculating the C-index in this study is right censoring, including patients who were lost to follow-up (a total of 670 patients were included, 18 were lost to follow-up) and patients who were still alive by the time of follow-up. We defined the survival time of patients who were lost to follow-up as the time from surgery to the last follow-up time, and the survival time of patients who were still alive in the end was defined as the time from surgery to the database cut-off time. This study uses the C-index software package of R software to calculate the C-index, which provides an inverse of the probability of censoring weighted estimate of the concordance probability to adjust for right censoring. We have supplemented the 'Statistical analysis' part of the 'Materials and Methods' section in the revised manuscript.

REVIEWERS' COMMENTS:

Reviewer #1 (Remarks to the Author):

The authors have conducted additional assays and data analysis to factor in tumor MSI status and EBV association for their 6 immune checkpoint marker assay. This is commendable given this was in response to this reviewer's comment and the bulk of literature to date supporting these molecular subtypes responding well to immune checkpoint inhibitors in the metastatic setting. Furthermore, MSI appears to carry favorable prognosis in the non-metastatic setting, and EBV may carry the same association.

Major comment:

The authors should provide some comment on the relatively larger proportion of MSI-H patients detected (25%) in comparison to other post-hoc analyses of large adjuvant trials to date reporting a range of 6-8%. Given the concordance of MMR IHC with MSI determination by traditional PCR methods, I believe this approach was reasonable but should not be responsible for potentially a high proportion of "false-positive" MSI-H cases. This also raises the question if false-positive MSI-H cases may have led to their surprising but provocative observation if can be externally validated for their ISS-GC score identifying a proportion of MSI-H gastric cancers deriving benefit from adjuvant chemotherapy. The authors should also provide some comment on the surprising contrast with other reports in the literature to date where assignment to adjuvant or perioperative chemotherapy in MSI-H gastric cancer patients either has no positive influence or possibly even detrimental survival outcomes.

Minor comment:

In the Discussion section, the authors introduced a new narrative discussing FDA approval for pembrolizumab. It is accurate that MSI status is a companion diagnostic, however EBV status is not nor is it listed in NCCN treatment guidelines to determine EBV status for anti-PD-1 therapy appropriateness based on the limited literature to date. For MSS and EBV negative gastric cancers, PD-L1 status as per the 22C3 Combined Positive Score assay is still an approved companion diagnostic for pembrolizumab. This portion of the discussion should be revised for accuracy.

Peer review report by:

Joseph Chao, MD
Department of Medical Oncology and Therapeutics Research
City of Hope Comprehensive Cancer Center

Reviewer #2 (Remarks to the Author):

The authors substantially strengthened the manuscript by showing multi-color IF staining data for the cell of origin of the checkpoint markers, and by estimating the immune infiltration scores of 444 samples. I don't have further major comments, just a few minor suggestions:

1. Figure 4 is a nice demonstration of high checkpoint expressing cells are mainly tumor. Is there a rough estimation of the percentage of such samples across all the 444 patients? It will be a very interesting finding if most of the GC tumors strongly express these immune suppressive markers. The authors could address this point in the discussion.
2. The authors should provide the infiltration scores as a supplementary table, maybe as an additional column in Table "Tissue Microarray".
3. Are the immune infiltration scores associated with patient outcome? Most GC tumors are immune cold, but it will be interesting to see that some of the 'hot' tumors in this study to show

some survival advantages.

Reviewer #3 (Remarks to the Author):

All my comments have been addressed

Reviewer #1

The authors have conducted additional assays and data analysis to factor in tumor MSI status and EBV association for their 6 immune checkpoint marker assay. This is commendable given this was in response to this reviewer's comment and the bulk of literature to date supporting these molecular subtypes responding well to immune checkpoint inhibitors in the metastatic setting. Furthermore, MSI appears to carry favorable prognosis in the non-metastatic setting, and EBV may carry the same association.

Major comment:

The authors should provide some comment on the relatively larger proportion of MSI-H patients detected (25%) in comparison to other post-hoc analyses of large adjuvant trials to date reporting a range of 6-8%. Given the concordance of MMR IHC with MSI determination by traditional PCR methods, I believe this approach was reasonable but should not be responsible for potentially a high proportion of "false-positive" MSI-H cases. This also raises the question if false-positive MSI-H cases may have led to their surprising but provocative observation if can be externally validated for their ISS-GC score identifying a proportion of MSI-H gastric cancers deriving benefit from adjuvant chemotherapy. The authors should also provide some comment on the surprising contrast with other reports in the literature to date where assignment to adjuvant or perioperative chemotherapy in MSI-H gastric cancer patients either has no positive influence or possibly even detrimental survival outcomes.

Response: Thanks very much for the very valuable comments. According to reports in literatures, the sensitivity and specificity of IHC and PCR detection of MSI/MMR were equally high, and the detection results of the two methods were highly consistent¹. In our

study, there were tissue wax blocks reserved for patients for IHC detection, so we chose IHC method to detect the MSI status of 444 patients. The literatures show that the overall detection rate of MSI-H in gastric cancer was 6-30% among different regions and races²⁻¹¹. Based on the slightly higher MSI-H detection rate, we have the following comments: First, a plenty of documents citing the MSI status of gastric cancer patients in the TCGA database showed that the detection rate was 21.9%, which was approximate to ours¹²⁻¹⁵. Second, several studies have shown that the detection rate of MSI-H was related to the age of patients¹⁶⁻¹⁸. Among the patients with MSI-H, the patients over 70 years old had the highest proportion, with statistical significance. Among the patients in our center, patients elder than 70 accounted for 27.9%, so the high detection rate of MSI-H may be related to the ages of patients. Third, it has been reported that the detection rate of MSI-H differs in different regions, for example, 7% in the United States, 15% in Mexico, and 50% in South Korea¹⁹. Therefore, MSI status may be related to race. It should be one of the reasons for the higher MSI-H detection rate in our center. Lastly, we selected 444 gastric cancer patients randomly in our center from 2010 to 2014. It only represented the MSI-H ratio of a single center, but not the overall ratio of multiple centers. Therefore, patients with multiple centers in different regions and a larger sample size would need to clarify the overall MSI-H detection rate of gastric cancer patients. We have added this explanation to the 'discussion' section in the revised manuscript.

In addition, it has been reported in literatures that adjuvant or perioperative chemotherapy has no significant positive effect on MSI-H patients, but our research aims to use the ISS_{GC} to distinguish gastric cancer patients who benefit from chemotherapy from the patients with MSI-H status, which is not contrast with the existing research results.

References :

- 1 Zhang X, Li J, et al. Era of universal testing of microsatellite instability in colorectal cancer. *World J Gastrointest Oncol*, **5**, 9-12(2013).

2 M. J.M. van Velzena, S. Derksb, et al. MSI as a predictive factor for treatment outcome of gastroesophageal Adenocarcinoma, *Cancer Treatment Reviews*, **86**, (2020).

3 Kim HJ, Kim N, Choi YJ, et al. Clinicopathologic features of gastric cancer with synchronous and metachronous colorectal cancer in Korea: are microsatellite instability and p53 overexpression useful markers for predicting colorectal cancer in gastric cancer patients? *Gastric Cancer*, **19**, 798-807(2016).

4 Di Bartolomeo M, Morano F, Raimondi A, et al. Prognostic and Predictive Value of Microsatellite Instability, Inflammatory Reaction and PD-L1 in Gastric Cancer Patients Treated with Either Adjuvant 5-FU/LV or Sequential FOLFIRI Followed by Cisplatin and Docetaxel: A Translational Analysis from the ITACA-S Trial, *Oncologist*, **25**, 19-47 (2019).

5 Ottini L, Falchetti M, Saieva C, et al. MRE11 expression is impaired in gastric cancer with microsatellite instability, *Carcinogenesis*, **25**, 2337-2343(2004).

6 Fang WL, Chen MH, Huang KH, et al. The Clinicopathological Features and Genetic Mutations in Gastric Cancer Patients According to EMAST and MSI Status. *Cancers (Basel)*, **12**, 551(2020).

7 Park J, Shin S, Yoo HM, et al. Evaluation of the Three Customized MSI Panels to Improve the Detection of Microsatellite Instability in Gastric Cancer, *Clin. Lab.* **63**, 705-716(2017).

8 Martinez-Ciarpaglini C, Fleitas-Kanonnikoff T, Gambardella V, et al. Assessing molecular subtypes of gastric cancer: microsatellite unstable and Epstein-Barr virus subtypes. *Methods for detection and clinical and pathological implications*, *ESMO Open*, **4**, (2019).

9 Arai T, Sakurai U, Sawabe M, et al. Frequent microsatellite instability in papillary and solid-type, poorly differentiated adenocarcinomas of the stomach, *Gastric Cancer*, **16**, 505-512(2013).

10 Kohlruss M, Grosser B, Krenauer M, et al. Prognostic implication of molecular subtypes and response to neoadjuvant chemotherapy in 760 gastric carcinomas: role of Epstein Barr virus infection and high and low microsatellite instability, *J Pathol Clin Res*, **5**, 227-239(2019).

11 Marrelli Daniele, Polom Karol, Pascale Valeria, et al. Strong Prognostic Value of Microsatellite Instability in Intestinal Type Non-cardia Gastric Cancer, *Ann. Surg. Oncol.* **23**, 943-950(2016).

12 Cortes-Ciriano Isidro, Lee Sejoon, Park Woong-Yang, et al. A molecular portrait of microsatellite instability across multiple cancers, *Nat Commun*, **8**, 15180(2017).

13 Germano Giovanni, Amirouchene-Angelozzi Nabil, Rospo Giuseppe, et al. The Clinical Impact of the Genomic Landscape of Mismatch Repair-Deficient Cancers, *Cancer Discov*, **8**, 1518-1528(2018).

14 Hause Ronald J, Pritchard Colin C, Shendure Jay, et al. Classification and characterization of microsatellite instability across 18 cancer types, *Nat. Med*, **22**, 1342-1350(2016).

15 Cancer Genome Atlas Research Network, Comprehensive molecular characterization of gastric adenocarcinoma, *Nature*, **513**, 202-209(2014).

16 Polom Karol, Marrelli Daniele, Roviello Giandomenico, et al. Molecular key to understand the gastric cancer biology in elderly patients-The role of microsatellite instability, *J Surg Oncol*, **115**, 344-350(2017).

17 Carolina Martinez-Ciarpaglini, Tania Fleitas-Kanonnikoff, et al. Assessing molecular subtypes of gastric cancer: microsatellite unstable and Epstein-Barr virus subtypes. Methods for detection and clinical and pathological implications, *ESMO Open*, **27**, (2018).

18 K. Polom, L. Marano, D. Marrelli, et al. Meta-analysis of microsatellite instability in relation to clinicopathological characteristics and overall survival in gastric cancer, *Bjs*, (2017).

19 Sepulveda A R, Santos A C, Yamaoka Y, et al. Marked differences in the frequency of microsatellite instability in gastric cancer from different countries, *Am. J. Gastroenterol*, **94**, 3034-3038(1999).

Minor comment:

In the Discussion section, the authors introduced a new narrative discussing FDA approval for pembrolizumab. It is accurate that MSI status is a companion diagnostic, however EBV status is not nor is it listed in NCCN treatment guidelines to determine EBV status for anti-PD-1 therapy appropriateness based on the limited literature to date. For MSS and EBV negative gastric cancers, PD-L1 status as per the 22C3 Combined Positive

Score assay is still an approved companion diagnostic for pembrolizumab. This portion of the discussion should be revised for accuracy.

Response: Thanks very much for the extremely valuable comments. We have revised the statements related to the FDA approval for pembrolizumab in the discussion section to make it more rigorous and accurate. The revised part of discussion was as follows: For example, the pembrolizumab has been approved by FDA for the treatment of patients with unresectable or metastatic and MSI-H GCs. Furthermore, some studies also showed that EBV-positive GCs were effective for the treatment of Avelumab and other immune checkpoint drugs ¹⁻³.

References:

1 Panda A, Mehnert J M, Hirshfield K M, et al. Immune activation and benefit from avelumab in EBV-positive gastric cancer, *J Natl Cancer Inst*, **110**, 316-320(2018).

2 Sho Sasaki, Jun Nishikawa, et al. EBV-associated gastric cancer evades T-cell immunity by PD-1/PD-L1 interactions. *Gastric Cancer*, **22**, 486-496(2019).

3 Sun YoungKim, CharnyPark, et al. Deregulation of Immune Response Genes in Patients With Epstein-Barr Virus-Associated Gastric Cancer and Outcomes, *Gastroenterology*, **148**, 137-147(2015).

Reviewer #2

The authors substantially strengthened the manuscript by showing multi-color IF staining data for the cell of origin of the checkpoint markers, and by estimating the immune infiltration scores of 444 samples. I don't have further major comments, just a few minor suggestions:

1. Figure 4 is a nice demonstration of high checkpoint expressing cells are mainly tumor. Is there a rough estimation of the percentage of such samples across all the 444 patients?

It will be a very interesting finding if most of the GC tumors strongly express these immune suppressive markers. The authors could address this point in the discussion.

Response: Thanks very much for your extremely suggestive comments. In this study, we have performed multi-colour immunofluorescence staining analysis in 135 gastric cancer tissues. Approximately 90% of tissues show that NECTIN2, CEACAM1, HMGB1, SIGLEC6 and CD15 were expressed in tumor cells. CD44 was expressed in tumor cells in about 70% of tissues. We have added this explanation to the 'discussion' section in the revised manuscript.

2. The authors should provide the infiltration scores as a supplementary table, maybe as an additional column in Table "Tissue Microarray".

Response: Thank you very much for your comments. We have provided the infiltration scores as the columns named 'CD3', 'CD8' and 'TIL' in the raw data of internal excel. We have uploaded all the raw data to the Springer Nature database. The supplemental figure 11 (Heat map) also presented the infiltration scores of 444 GC patients.

3. Are the immune infiltration scores associated with patient outcome? Most GC tumors are immune cold, but it will be interesting to see that some of the 'hot' tumors in this study to show some survival advantages.

Response: Thanks very much for the very suggestive comments. To evaluate the prognostic value of the immune infiltration score, Kaplan-Meier analysis and stratification analysis were performed by ISS_{GC} . As shown in Figure 1A (also as Supplemental Figure S16A in the revised manuscript), although the overall immune infiltration score did not reach statistical significance, it still showed a clear trend that higher immune infiltration score indicated a better prognosis. And as shown in Figures 1B and 1C (also as Supplemental Figures S16B and 16C in the revised manuscript), the immune infiltration score was able to distinguish patients with poor prognosis by the low- ISS_{GC} stratification. But it can not predict the prognosis through the immune infiltration score for the patients

with high-ISS_{GC}. We have added this explanation to the 'discussion' section in the revised manuscript.

Figure1 A: Overall survival based on the immune infiltration score in 444 GC patients. B,C: Overall survival based on the immune infiltration score in GC patients after ISS_{GC} stratification.